# Microstructural control suppresses thermal activation of electron transport at room temperature in polymer transistors

Alessandro Luzio[1], Fritz Nübling[2], Jaime Martin [3,4], Daniele Fazzi[5], Philipp Selter[6], Eliot Gann[7,8,11], Christopher R. McNeill [7], Martin Brinkmann[9], Michael Ryan Hansen [6], Natalie Stingelin [10], Michael Sommer[2] & Mario Caironi [1]

Recent demonstrations of inverted thermal activation of charge mobility in polymer field-effect transistors have excited the interest in transport regimes not limited by thermal barriers. However, rationalization of the limiting factors to access such regimes is still lacking. An improved understanding in this area is critical for development of new materials, establishing processing guidelines, and broadening of the range of applications. Here we show that precise processing of a diketopyrrolopyrrole-tetrafluorobenzene-based electron transporting copolymer results in single crystal-like and voltage-independent mobility with vanishing activation energy above 280 K. Key factors are uniaxial chain alignment and thermal annealing at temperatures within the melting endotherm of films. Experimental and computational evidences converge toward a picture of electrons being delocalized within crystalline domains of increased size. Residual energy barriers introduced by disordered regions are bypassed in the direction of molecular alignment by a more efficient interconnection of the ordered domains following the annealing process.

[1] Center for Nano Science and Technology@PoliMi, Istituto Italiano di Tecnologia, via Giovanni Pascoli 70/3, Milan 20133, Italy. [2] Technische Universität Chemnitz, Polymerchemie, Straße der Nationen 62, 09111 Chemnitz, Germany. [3] POLYMAT, University of the Basque Country UPV/EHU, Avenida de Tolosa 72, 20018 Donostia-San, Sebastián, Spain. [4] Ikerbasque, Basque Foundation for Science, 48013 Bilbao, Spain. [5] Institut für Physikalische Chemie, Department Chemie, Universität zu Köln, Luxemburger Str. 116, D - 50939 Köln, Germany. [6] Institut für Physikalische Chemie, Westfälische Wilhelms-Universität, Corrensstraße 28, 48149 Münster, Germany. [7] Materials Science and Engineering, Monash Univeristy, Clayton, VIC 3800, Australia. [8] Australian Synchrotron, ANSTO, Clatyon, VIC 3168, Australia. [9] Institut Charles Sadron, CNRS, Université de Strasbourg, 23 rue du Loess, BP 84047, Cedex 2 67034 Strasbourg, France. [10] School of Materials Science and Engineering, Georgia Institute of Technology, 771 Ferst Drive, Atlanta 33022 GA, USA. [11]Present address: National Institute of Standards and Technology, Gaithersburg, MD 20899, USA. Correspondence and requests for materials should be addressed to M.S. (email: michael.sommer@chemie.tu-chemnitz.de) or to M.C. (email: mario.caironi@iit.it)

Semiconducting polymers with ideal, band-like transport are strongly desired to meet the requirements for a vast range of applications in the field of flexible, large-area electronics[1,2], including wearable, portable and distributed sensing, monitoring and actuating devices[3–5]. This need derives from a fundamental limit to the maximum charge mobility in temperature activated transport regimes, which is generally assumed to be <1 cm$^2$/Vs for hopping[6,7]. Such performance cannot meet the increasingly stringent requirements for emerging thin film technologies such as high resolution backplanes, full color displays and sensors networks, among others.

A vast library of high performance donor-acceptor copolymers has been developed in recent years, making solution processed conjugated polymers yet more appealing for large-area and flexible electronic applications[4,8–13]. Among them, materials with field-effect electron mobilities exceeding 1 cm$^2$/Vs are no longer isolated examples[13], demonstrating that n-type polymer field-effect transistors (FETs) are catching up their p-type counterparts[12]. Band-like transport in organic materials is now well established in the case of small molecules, both for single crystals[14–17] and more recently for thin films[18,19]. Likely because of the intrinsically higher degree of disorder in polymer semiconductors, very few cases of inverted temperature activated transport have been demonstrated for p-type polymer devices[20–23], and only a single one for n-type devices[24]. The observation of such inverted temperature activation has been so far assigned either to a specific chemical structure[24], to processing conditions[20], or to the degree of backbone alignment[21–23]. It is indeed of utmost importance to clarify the conditions necessary to observe the transition from thermal activation towards temperature independent and band-like transport in polymer FETs.

Here we report the systematic investigation of structure-function relationships of a solution-processable polymer comprising alternating dithienyldiketopyrrolopyrrole (ThDPPTh) and tetrafluorobenzene (F4) units, referred to as PThDPPThF4[25]. PThDPPThF4 used herein is synthesized by direct arylation polycondensation using a simple four-step protocol, and is free of homocoupling defects[26]. We demonstrate that the combination of uniaxial alignment and thermal annealing within the temperature range where partial melting of crystals occurs, gives access to efficient temperature-independent electron transport above 280 K, at the boundary between temperature-activated and band-like regimes. The transport properties between these two regimes can be controlled by distinct thermal annealing protocols below, in close proximity to, and above the melting temperature ($T_m$) of the material. Annealing at a temperature at which the smaller crystals melt, but the larger ones are maintained, allows for crystal thickening while preserving uniaxial chain alignment. Thus, a gate bias-independent electron mobility of 3 cm$^2$/Vs is achieved. Using detailed thermal, morphological, opto-electronic, and theoretical methods, we are able to assign the origin of the improved transport properties to the synergy of improved order within the crystalline domains, increased interconnectivity of such domains, and a significantly reduced contribution of the disordered regions. Our findings offer practical processing guidelines for microstructure engineering in high mobility polymer thin films beyond thermally limited charge transport.

## Results

### Thermal characterization of PThDPPThF4 thin films.

Thermal annealing is a widely employed strategy to enhance the transport properties of semiconducting polymer thin films via rearrangement of microstructure. Finding the optimal annealing temperature is usually either empirical or based on the phase behavior deduced from differential scanning calorimetry (DSC) performed on bulk

samples. However, it is well-known that the phase behavior of bulk polymer materials often differs from that of thin films[27], which are typically employed in FETs. Furthermore, in the spin casting process the polymer rapidly solidifies, giving rise to a kinetically trapped, non-equilibrium microstructure. Hence, the direct correlation between the phase behavior, including the thermal transitions, of spin coated thin films and standard DSC data is not always accurate. In order to assess the thermal behavior and the effect of thermal annealing on PThDPPThF4 (Fig. 1a, number-average molecular weight $M_n = 14\,\mathrm{kg\,mol^{-1}}$; dispersity, Đ = 3.7, size-exclusion chromatography in Supplementary Fig. 1, cyclic voltammetry in Supplementary Figs. 2 and 3, extracted LUMO values in Supplementary Table 1) thin films processed under the conditions employed in FET devices, we conducted fast scanning calorimetry (FSC) instead of standard DSC bulk experiments. This allows a reliable evaluation of the first heating curves and avoids further structural reorganization during heating owing to fast heating rates[28,29]. For a complete discussion of DSC vs. FSC see the Supplementary Figs. 4 and 5 and Supplementary Note 1. The FSC traces of the 1st and the 2nd heating and the 1st cooling scans of a spin cast, ~40 nm thin PThDPPThF4 film at a scan rate of 2000 °C s$^{-1}$ are shown in Fig. 1b. We associate the main endothermic peaks in heating scans with melting of the crystalline regions of PThDPPThF4. The melting temperatures ($T_m$) for the as-cast (1st heating scan) and the melt-crystallized (2nd heating scan) samples amount to 283 and 285 °C, respectively. Likewise, the main exothermic peak at 261 °C in the cooling trace corresponds to crystallization of PThDPPThF4. Because the 1st heating trace in Fig. 1b reflects the microstructure developed during spin coating, it can be used as a guide to judiciously select thin film annealing temperatures. Accordingly, the annealing temperatures $T_1$, $T_2$, and $T_3$ were selected as temperatures well-below, within and above the melting endotherm, respectively. At $T_3$, PThDPPThF4 is in a liquid state where molecular order is lost, and the final microstructure of the film at room temperature depends on the cooling kinetics. Conversely, PThDPPThF4 films are semicrystalline when annealed at $T_1$ and $T_2$ and thus the impact of annealing at those temperatures can be evaluated by calorimetry. To do so, we designed thermal protocols shown in Fig. 1c. $T_1 = 210\,°C$ and a $T_2 = 287\,°C$ were selected according to the data presented in Fig. 1b. Classical isothermal crystallization experiments in which films are quenched from the melt to a pre-selected crystallization temperature were not considered so as not to erase backbone alignment. The black traces in Fig. 1d correspond to the calorimetric signal recorded when heating an as-spun film up to the annealing temperature (stage I, Fig. 1c). The red and the blue traces correspond to the heating scans recorded immediately after the isothermal 5 min periods at $T_1$ and $T_2$, respectively (stage III). We find that annealing at $T_1$ has no impact on the melting behavior, as expected. On the contrary, annealing at $T_2$ provokes a marked shift of the melting endotherm towards higher temperatures with $T_m$ increasing by ~30 K with respect to $T_m$ of the pristine film and the film annealed at $T_1$. This increase in $T_m$ is rationalized in terms of thickening of the crystalline lamellae during annealing at $T_2$, in agreement with the Gibbs-Thomson theory[30]. Accordingly, in PThDPPThF4 films annealed at a temperature within the melting temperature range, the thinner crystalline lamellae melt, while the thicker lamellae remain solid and evolve towards thermodynamic equilibrium, which corresponds to fully chain extended crystals[31]. Therefore, during annealing at $T_2$, the thickness of the solid lamellae significantly increases, which results in the observed, notable increase of $T_m$. This behavior is common to all PThDPPThF4 samples analyzed, independently on chain length and molecular characteristics, as demonstrated in the Supplementary Information for samples of $M_n = 11$, 14 and 30 kg mol$^{-1}$ (Supplementary Fig. 6 and Supplementary Note 2). Further confirmation comes from

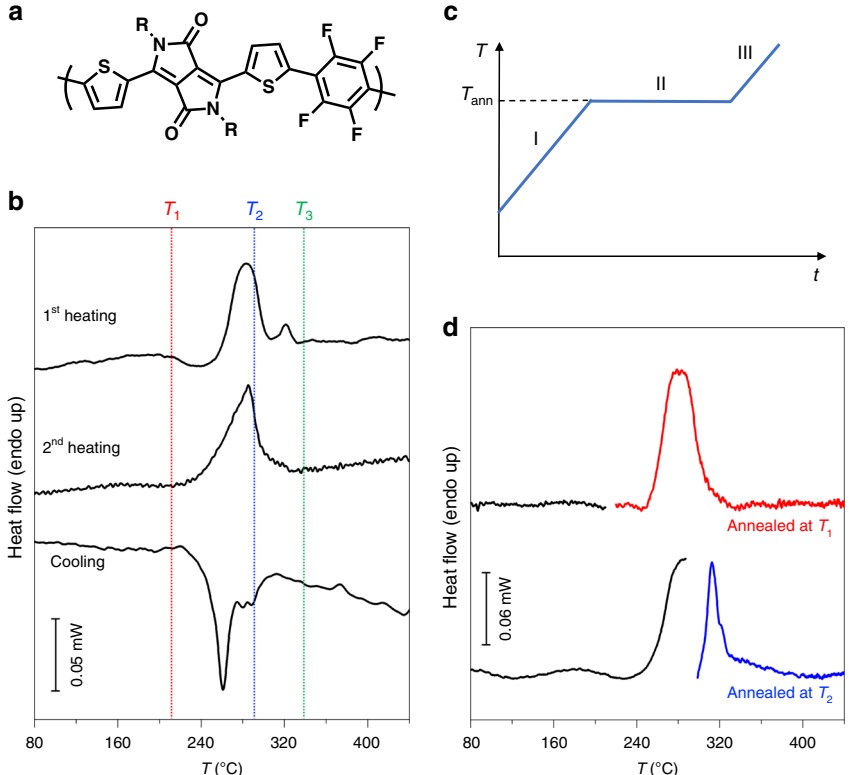

**Fig. 1** Fast scanning calorimetry data. **a** Molecular structure of PThDPPThF4. R = 2-octyldodecyl. **b** Fast scanning calorimetry (FSC) 1st heating, 2nd heating and cooling traces for a spin-cast thin film of PThDPPThF4 with $M_n$ = 14 kg mol$^{-1}$. The three temperatures $T_1$, $T_2$, and $T_3$ are below, within and above the melting endotherm, respectively and used throughout this study. **c** Temperature program of isothermal annealing: stage I is the heating ramp from 30 °C to the annealing temperature ($T_{ann}$) at a rate of 2000 °C s$^{-1}$, stage II is a 5 min annealing period at $T_{ann}$, and stage III corresponds to a scan from $T_{ann}$ to 450 °C at 2000 °C s$^{-1}$. **d** Heating traces using the isothermal annealing protocol shown in **c** with $T_1$ = 210 °C and $T_2$ = 287 °C. The black lines correspond to the calorimetric signal recorded as heating as spun samples up to the annealing temperatures, i.e. segment I in **c**. The red and the blue lines correspond to the heating scans recorded immediately after the 5-min-annealing steps at $T_1$ and $T_2$, respectively, i.e. segment III in **c**

measurements performed at lower scanning rates, using standard DSC and polarized optical microscopy and spectroscopy (see Supplementary Figs. 5, 7–9 and Supplementary Note 3)[29,32].

**Optical characterization of aligned films**. A strong enhancement of charge mobility in polymer semiconductors can be obtained with uniaxial backbone alignment, inducing transport anisotropy and favoring improved transport properties along the alignment direction[33]. To exploit such behavior, we have realized uniaxially aligned films using either off-center spin coating (Fig. 2a), or wired-bar coating, according to a methodology based on the exploitation of marginal solvents and directional flow during deposition[34,35]. These two techniques produce similarly aligned and microstructured films.

Figure 2b shows UV-Vis spectra of aligned films after annealing at $T_1$, $T_2$, and $T_3$. All spectra feature a low-energy (namely, charge transfer or CT band) and a high-energy (not shown) band. While significant spectral changes between films annealed at $T_1$ and $T_2$ cannot be observed, a strong hypsochromic shift occurs after melt-annealing at $T_3$. Spectra of films annealed at $T_1$ and $T_2$ show three contributions, two bands at low energy (≈1.60 and ≈1.77 eV) assigned to 0–0 and 0–1 transitions within PThDPPThF4 aggregates, and a shoulder peaking at ≈1.88 eV, which is assigned to disordered polymer chains[26]. Following this assignment, we can observe that the content of crystalline regions is similar for films annealed at $T_1$ and $T_2$. Upon melt annealing at $T_3$ and successive cooling, the thermal history of the film is erased. As a consequence, the band at 1.88 eV dominates the spectrum owing to an increased molecularly disordered structure.

Measurements of the dichroic ratio ($DR$) reveals optical anisotropy after annealing at $T_1$ and $T_2$ of $DR ≈ 2$ for both cases, implying preferential alignment of polymer chains along the radial direction induced during off-center spin coating. It is important to emphasize that uniaxial alignment is maintained upon annealing at $T_2$, which is a consequence of the thicker crystals acting as nuclei that dictate chain and crystal orientation. In contrast, melting at $T_3$ followed by cooling to room temperature entirely erases any preferred backbone alignment.

**Microstructural characterization of aligned films**. Two-dimensional grazing incidence wide-angle X-ray scattering (GIWAXS) was employed to gain information on how these annealing processes modify morphology and coherence lengths. d-spacings and coherence lengths from GIWAXS patterns are summarized in Supplementary Table 2 (for GIWAXS patterns see Supplementary Figs. 10–13 and Supplementary Note 4). The sample annealed at $T_1$ shows a semicrystalline morphology with pronounced edge-on orientation of chains (as evaluated through Herman's S-parameter, $S = 0.49$, Supplementary Fig. 12 and Supplementary Table 2). The main chain-side chain separation (100 reflection) and π-stacking distance (010 reflection) are 2.04 nm and 0.37 nm, respectively. Upon annealing at $T_2$, both the main chain-side chain separation distance and the π-stacking distances increase to 2.11 and 0.38 nm, respectively. A population of face-on crystallites appears, which shifts the $S$-value to $S = 0.32$. The coherence lengths in both stacking directions improve considerably compared to films annealed at $T_1$: from 13.7 nm to

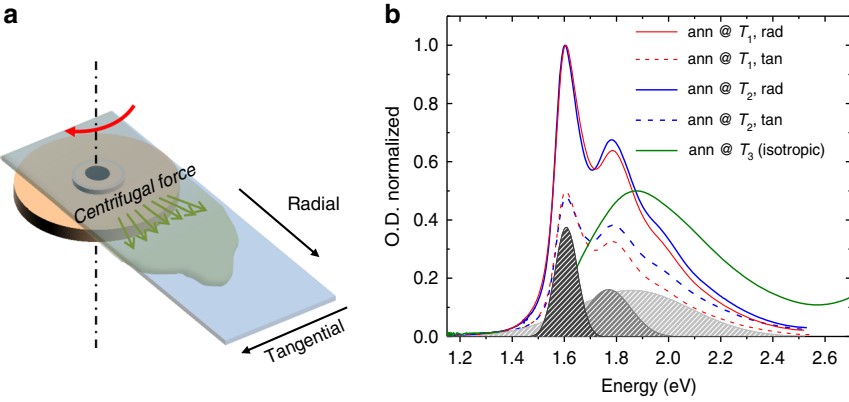

**Fig. 2** Uniaxial alignment and optical anisotropy. **a** Schematic of the off-center spin coating deposition method; **b** polarized UV-Vis spectra of films aligned using off-center spin coating and subsequently annealed at $T_1$, $T_2$, and $T_3$. The grey filled Gaussians result from the fitting of the spectrum of the film annealed at $T_1$

16.45 nm for the (100) and from 5.37 nm to 7.28 nm for the (010) reflection. This is a clear indication of an increased crystal dimension upon annealing at $T_2$, in agreement with FSC data. Melt-annealing at $T_3$ results in a major loss of crystalline scattering features, in agreement with optical characterization (Fig. 2). In the sample annealed at $T_3$, mainly diffuse scattering and weak (100) and (010) reflections are observed. The π-stacking distance of a minor residual crystalline fraction is still seen but increases to ~0.44 nm at a reduced coherence length of 1.36 nm. Thus, $T_3$ annealed films are weakly crystalline and characterized by much smaller and more disordered crystallites.

Near-edge X-ray absorption fine structure (NEXAFS) spectroscopy measurements on aligned films support the view of a strong in-plane alignment of polymer chains at the surface of films annealed at $T_1$ and $T_2$ (Supplementary Figs. 14 and 15 and Supplementary Note 5). Transmission electron microscopy (TEM) analysis of films annealed at $T_1$ and $T_2$ (Fig. 3a–d) shows in-plane π-stacking of mostly edge-on oriented crystallites, and sharper π-stacking upon $T_2$ annealing, indicating increased crystalline order, in agreement with GIWAXS analysis. The samples annealed at $T_1$ show some reflections along the chain direction at 2.7 and 2.3 Å, suggesting periodic ordering of the monomeric units. In the sample annealed at $T_2$, reflections along the chain direction disappear. Both TEM and also atomic force microscopy (AFM) indicate a fibrillary (i.e. fibrils grown parallel to chain axes) to lamellar (i.e. lamellae grown parallel to π-stacking direction) morphological transition from $T_1$ to $T_2$ annealed films (Fig. 3a, c, e, f), with long periods of 28–29 nm (Fig. 3b, d).

**Local packing from solid-state NMR.** To gain detailed information about temperature-dependent polymer chain conformation and local molecular packing of PThDPPThF4, we performed solid-state $^1$H and $^{19}$F magic-angle spinning (MAS) NMR experiments on samples annealed at different temperatures. These experiments rely on the reintroduction of the homonuclear $^1$H-$^1$H and $^{19}$F-$^{19}$F dipolar coupling via 2D double-quantum single-quantum (DQ-SQ) NMR correlation experiments[36], providing molecular information about spatial proximities and chain conformations[37,38]. In addition, 2D $^1$H-$^{13}$C heteronuclear correlation (HETCOR) spectroscopy experiments were performed. Like the $^1$H and $^{19}$F MAS NMR techniques, these experiments utilize the heteronuclear direct dipolar coupling to transfer magnetization directly through space from $^1$H to $^{13}$C, thereby offering the potential to establish through-space interchain correlations.

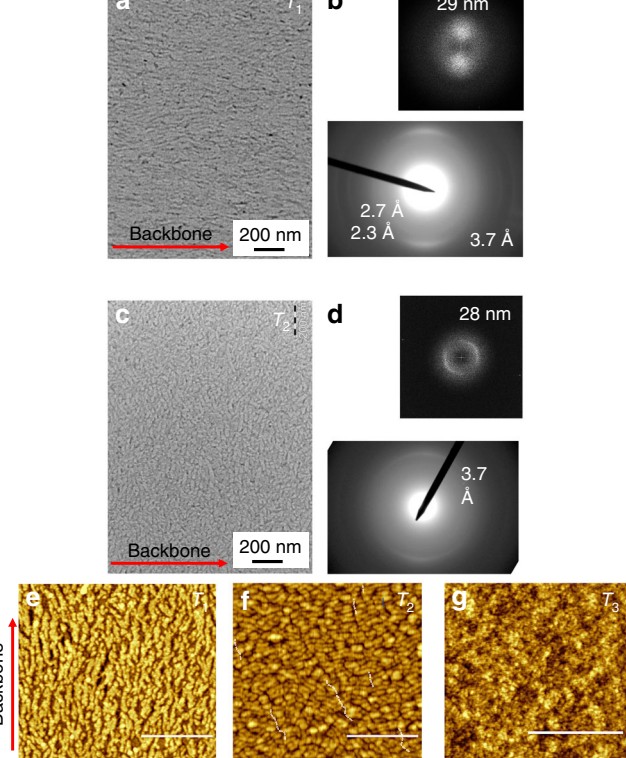

**Fig. 3** Microscopic characterization. TEM analysis and electron diffraction of $T_1$ (**a**, **b**) and $T_2$ (**c**, **d**) annealed films; **e–g** AFM topography images of oriented films annealed at $T_1$ ($R_{r.m.s.} = 1.2$ nm), $T_2$ ($R_{r.m.s.} = 0.62$ nm), and $T_3$ ($R_{r.m.s.} = 0.54$ nm). Scale bar for AFM images: 500 nm. Elongated fibrils can be observed on the surface of films annealed at $T_1$ (**e**), clearly aligned along the radial direction of spin. Upon annealing at $T_2$ (**f**), fibrils transform into crystalline lamellae of ~30 nm thickness, which apparently are oriented perpendicular to the radial direction of the spin (i.e. chains being aligned radially to the spin), in agreement with TEM analysis. Backbone direction, however, does not change upon annealing at $T_2$. After solidification from $T_3$ (**g**), the top surface is composed of small domains that lack any evidence of an ordered structure

On the basis of the observations from 2D $^1$H-$^1$H DQ-SQ and $^{19}$F-$^{19}$F DQ-SQ NMR correlation spectra (Supplementary Figs. 16–24, Supplementary Notes 6–8), it can be concluded that the disordered phase of the samples is characterized by non-

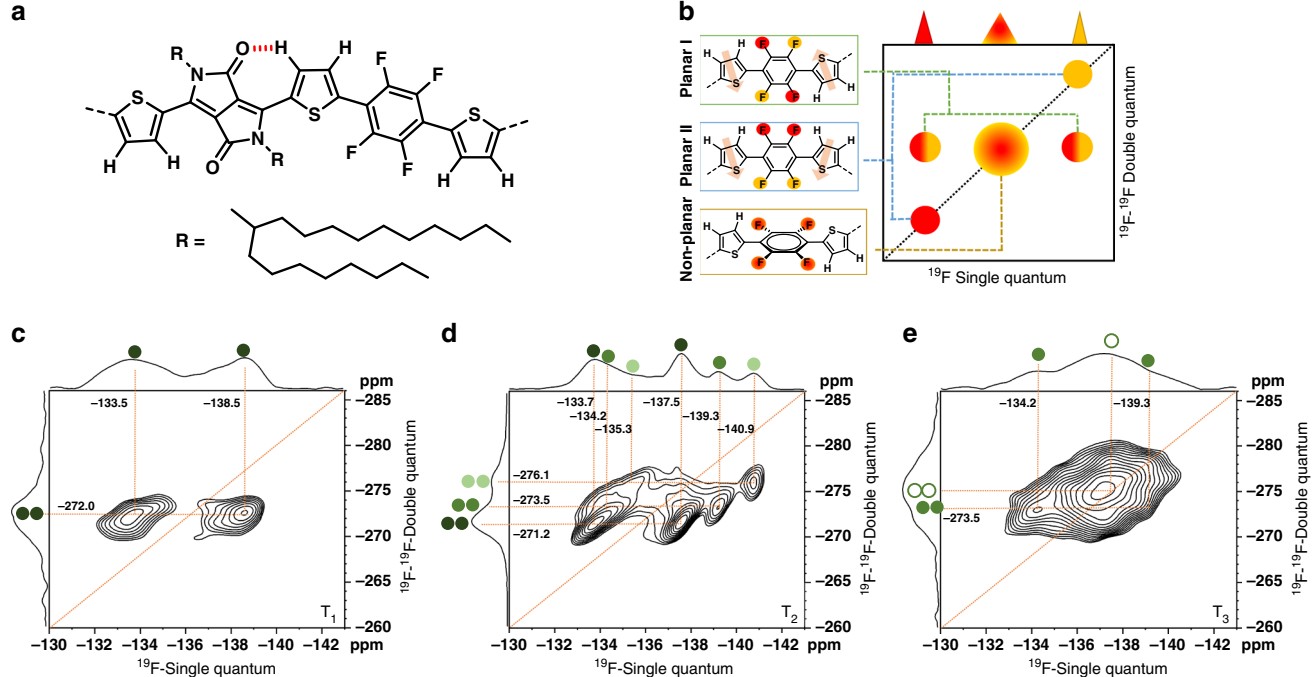

**Fig. 4** Solid-state NMR data. Results from $^{19}F$-$^{19}F$ solid-state NMR. **a** Conformation of the polymer backbone inside the ordered domains after annealing at $T_1$. **b** schematic representation for the interpretation of the $^{19}F$-$^{19}F$ DQ-SQ correlation spectra shown in **c–e**: while only one broad pair of cross-correlations is observed after annealing at $T_1$, three new cross-correlation pairs with significantly narrower lines are observed after annealing at $T_2$. Annealing at $T_3$ leads to mostly non-planar conformations around the F4 monomer

planar Th-F4-Th conformations and less ordered aliphatic side chains. Conversely, a strongly planarized molecular conformation is observed within the ordered domains of the samples annealed at $T_1$ and $T_2$.

Annealing at $T_1$ leads to a single stacking configuration for PThDPPThF4 in its ordered domains, with one prevalent polymer conformation as schematically illustrated in Fig. 4a. This conformation includes an intramolecular hydrogen bond between the DPP and neighboring Th group, as well as a co-planar 'anti' conformation of the thiophene groups surrounding the F4 units. Combining $^1H$-$^1H$ DQ-SQ and $^{19}F$-$^{19}F$ DQ-SQ correlation data with insights from $^1H$-$^{13}C$ HETCOR data (Supplementary Fig. 24), we propose a packing model for the polymer in the ordered domain, with the DPP units of one chain being placed above/below the F4 units of neighboring polymer chains. While this arrangement places an electron-deficient DPP unit above another electron-deficient F4 unit, it maximizes the distance between the sterically demanding aliphatic side chains, suggesting that the formation of this packing mode is at least partially driven by a reduction in steric hindrance. Taken together, this leads to the packing model I shown in Fig. 5b, corresponding to a shifted packing of the PThDPPThF4 main chains, where the DPP units are sandwiched in between F4 units from above and below.

Going to the higher annealing temperature $T_2$, no significant changes in conformation in the ordered domains are observed from both $^1H$ and $^{19}F$ NMR spectra. However, $^1H$-$^1H$ DQ-SQ correlation data indicates an increase in the order of the aliphatic side chains (Supplementary Fig. 18), while $^{19}F$-$^{19}F$ DQ-SQ correlation data shows the occurrence of three distinct cross-correlation peaks, with different chemical shifts compared to the correlations observed for $T_1$ (Fig. 4c–e). Since any conformational change around the F4 unit should result in either the occurrence of one or two auto-correlations (Fig. 4b), the emergence of three distinct cross-correlation peaks clearly indicates the formation of

at least two new packing modes, henceforth referred to as packing mode II and III (Fig. 5c, d). The structure of these two packing modes can be deduced by combining the results of $^1H$ and $^{19}F$ MAS NMR with further $^1H$-$^{13}C$ experiments, establishing a picture where packing modes II and III emerge from mode I via a slip of the polymer chains along their long axes, placing the DPP and F4 units now above neighboring Th units. This places the electron-deficient groups above and below electron-rich Th units, likely resulting in a lower overall energy. Sliding of the polymer backbones is possibly facilitated by an increase in order of the aliphatic chains, resulting in less steric demand in the long-axis direction of the polymer backbone and, as evident from GIWAXS, in an increased main chain-side chain separation distance with respect to $T_1$.

Since for every DPP and F4 unit two Th units are present in the polymer chain, the slip can occur in two directions (Fig. 5e) leading to the formation of either packing mode II or III. Interestingly, packing mode II retains point reflection symmetry in the F4 unit, resulting in two of the $^{19}F$ sites at the F4 unit being equivalent, while this symmetry is no longer present in mode III, where all four $^{19}F$ sites at the F4 unit are inequivalent.

Heating above the melting temperature at $T_3$ breaks a large fraction of the co-planar 'anti' conformations of the F4 group with respect to the neighboring Th units, increasing dihedral angles and leading to a packing structure of PThDPPThF4 dominated by random intra-chain conformations.

**FETs characterization at room temperature**. We fabricated bottom-contact, top-gate FETs using uniaxially aligned PThDPPThF4 thin films and a ≈550 nm thick PMMA dielectric layer. For uniaxial alignment, we employed alternatively off-center spin coating, as for optical and structural characterization samples, and wired-bar coating, a fast, large-area and scalable directional printing process[35].

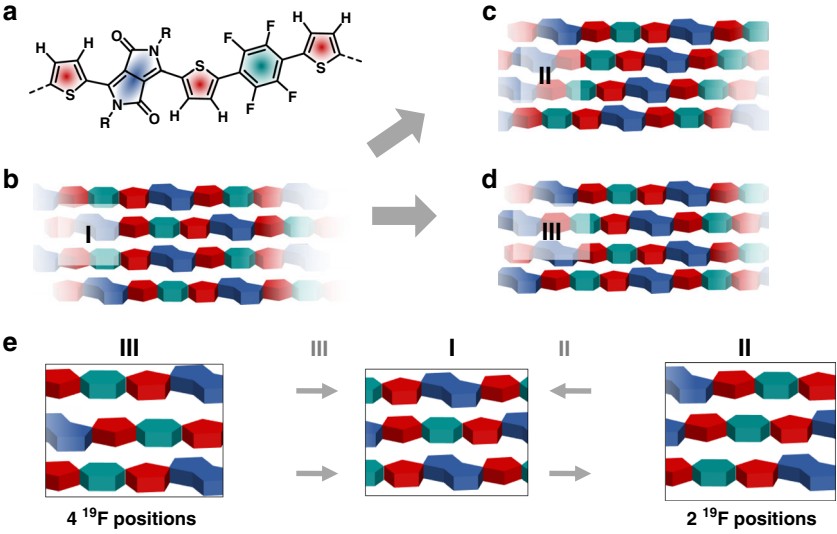

**Fig. 5 Molecular packing.** Schematic representation of local packing from solid-state NMR. **a** The predominant conformation in the ordered domains is characterized by an intra-molecular hydrogen bond and the 'anti' configuration of the thiophene monomers next to the F4 unit; **b** packing model I for the ordered domains after annealing at $T_1$, which upon annealing at $T_2$ transforms into packing models II **c** and III **d**. The three models are connected via a slipping motion involving the both neighboring chains moving in either the same or opposite directions **e**

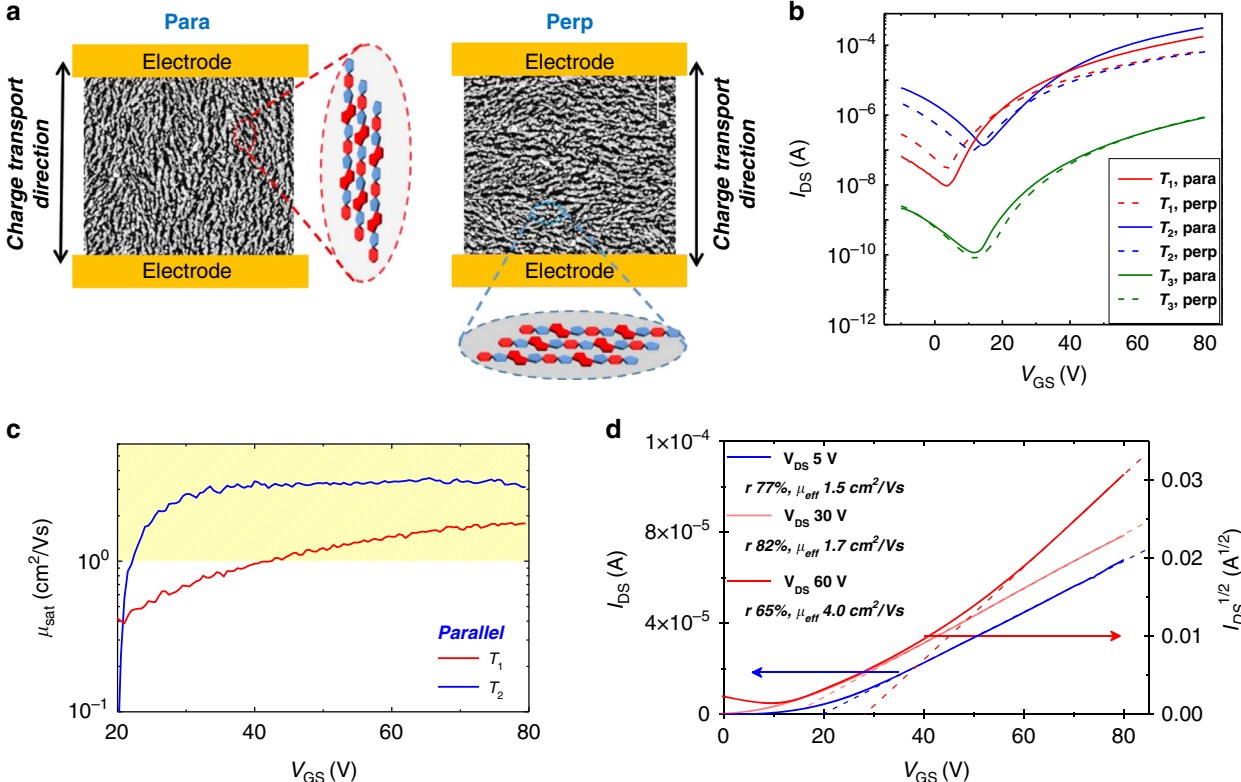

**Fig. 6 Electrical data of FETs.** FET characterization of PThDPPThF4. **a** Sketch of source and drain pattern configuration parallel (para) and perpendicular (perp), allowing to probe transport parallel and perpendicular to the chain direction, respectively; **b** transfer curves ($W = 1000\,\mu m$ $L = 80\,\mu m$), deposited using off-center spin coating, annealed at $T_1$, $T_2$ and $T_3$ and probed parallel and perpendicular to chain direction; **c** saturation mobility values ($\mu_{sat}$) vs. $V_{GS}$ plot of films annealed at $T_1$, $T_2$ probed parallel to chain direction; **d** $I_{DS}$ vs. $V_{GS}$ and $I_{DS}^{1/2}$ vs. $V_{GS}$ transfer curves measured at $V_{DS} = 5\,V$, $V_{DS} = 30\,V$, and $V_{DS} = 60\,V$, for devices annealed at $T_2$ and including an injection interlayer. Current values are recorded parallel to the alignment direction. It is worth noting that $\mu_{eff}$ of $\approx 1.5\,cm^2/Vs$ is obtained already in the linear regime ($V_{DS} = 5\,V$), as a result of a high reliability factor of $\approx 77\%$. Such a high $r$ value indicates FET behavior not far from ideality. In the present case the reliability factor is mostly limited by the non-zero threshold voltage. As a confirmation, at the onset of the saturation regime ($V_{DS} = 30\,V$), an $r$ up to ~82% is extracted, due to a reduced threshold voltage of $V_{Th} = 14\,V$. Moreover, a notable $\mu_{eff}$ value of $\approx 4\,cm^2/Vs$ is obtained at $V_{DS} = 60\,V$ ($r = 65\%$, $\mu_{sat} = 6.1\,cm^2/Vs$). FET parameters of curves in **d**: $W = 1000\,\mu m$; $L = 100\,\mu m$; $C_i = 5.7\,nF/cm^2$

To avoid extrinsic phenomena such as contact resistance and current-induced self-heating (Joule effect) leading to a questionable extraction of intrinsic transport parameters[7,39–41], FETs with a large channel length (from 80 to 100 µm) were mainly employed for mobility extraction.

The transport properties were investigated both with backbone alignment orientation parallel (Fig. 6a, para source & drain configuration) and perpendicular (Fig. 6a, perp source & drain configuration) to the charge transport direction (as defined by the orientation of electrical contacts). In Fig. 6b, representative transfer curves of FETs with 80 µm channel length are shown (output curves are reported in Supplementary Figs. 25 and 26). All devices exhibit typical ambipolar V-shaped curves with a clear prevalence for $n$-type current modulation and a weak p-type current appearing below 10 V. Gate leakage current is always much lower than drain-source current in all the explored voltage range (Supplementary Figs. 25a and 26a). Backbone alignment results in charge transport anisotropy with improved transport properties along the direction of polymer backbones (para, Fig. 6b). In films annealed at $T_1$, a 2.6 × higher source to drain current ($I_{DS}$) is recorded with respect to perp configuration at the maximum $V_{GS}$ employed in saturation regime. In films annealed at $T_2$, the $I_{DS,para}/I_{DS,perp}$ ratio increases to ≈5, owing to an improvement of transport only in the parallel direction. In films processed at $T_3$, transport anisotropy is no longer present, in agreement with the loss of uniaxial alignment, and a general drop of current is observed.

Apparent saturation mobility values were first extracted from the local slopes of $I_{DS}^{1/2}$ vs. $V_{GS}$ for the saturation regime and of $I_{DS}$ vs. $V_{GS}$ for the linear regime, according to the gradual channel approximation equations[42]. Focusing on the transport properties parallel to backbone alignment (Fig. 6c), for films annealed at $T_1$, gate voltage-dependent mobility values are extracted within the entire investigated range, i.e. up to $V_{GS} = 80$ V. At the maximum applied $V_{GS}$ the maximum apparent mobility extracted is ≈1.7 cm²/Vs. Such marked voltage dependence of mobility in a long channel polymer transistor where contact effects should be mitigated can be associated to charge density transport dependence in a broad density of states (DOS) and/or trap filling[43,44].

More interestingly, upon annealing at $T_2$, the apparent mobility value parallel to backbone alignment ($\mu_{para}$) rapidly saturates to a constant value at relatively low $V_{GS} > 30$ V, i.e. just 10 V above the threshold voltage $V_{Th}$ (Fig. 6c), denoting more ideal transport characteristics. Moreover, the electron mobility is found to exceed 1 cm²/Vs for $V_{GS}$ values very close to $V_{Th}$ (i.e. at the early onset of the device). To further reduce $V_{Th}$, we implemented an injection layer into the device (see details in Supplementary Fig. 27, and transfer curves in Fig. 6d), leading to a reduction of $V_{Th}$ by ~5 V with no variation of mobility values. This is indicative of a contact resistance-dominated turn-on voltage which may be further engineered and improved.

As a more reliable parameter and to take into account device non-idealities, we extracted the effective mobility $\mu_{eff}$ as the product between the apparent mobility and the measurement reliability factor $r$, as recently defined by Choi et al. (Supplementary Figs. 28 and 29 and Supplementary Table 3)[45]. Notable $\mu_{eff}$ values of ≈1.5 cm²/Vs in the linear regime of ≈4 cm²/Vs in saturation regime are obtained. It is worth highlighting that the high reliability values for films annealed at $T_2$ ($r$ up to ~82%, see Fig. 6d) compare well to those obtained with polymeric p-type counterparts and, remarkably, with small molecule thin films and single crystal based devices, leading to impressively high effective mobilities[46–49]. When films are subjected to a temperature above melting ($T_3$), the mobility drastically decreases to $10^{-2}$ cm²/Vs and a strong $V_{GS}$ dependence of the mobility is observed. Such

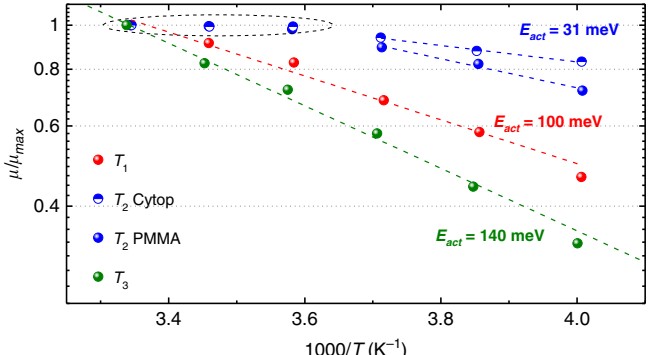

**Fig. 7** Mobility as a function of temperature. Comparison of $\mu_{para}/\mu_{max}$ vs. $1/T$ from the FETs of this work in the direction parallel to backbone alignment

electrical behavior is in agreement with increased microstructural/conformational disorder and consequently strong DOS broadening as suggested by the above optical and microstructural characterization.

**Temperature-dependent characterization of FETs**. It has been predicted that for charge mobility values exceeding 1 cm²/Vs, the transport regime must clearly deviate from a thermally activated mechanism, typically observed in most polymer semiconductors[6]. In order to get insight into the transport mechanism that is operative in aligned films of PThDPPThF4, we performed electrical characterization at variable temperature (Fig. 7). For annealing at $T_2$, the $\mu_{perp}$ vs. $1/T$ plot can be well described by an Arrhenius dependence over the entire temperature range, with an activation energy $E_a$ of ≈100 meV at any $V_{GS}$ employed for mobility extraction (Supplementary Fig. 30). Differently, at sufficiently high $V_{GS}$, $\mu_{para}$ is constant down to 280 K. This observation is reproducible in different devices and with different dielectrics (Supplementary Fig. 31) and reveals a temperature range in which transport is not thermally activated, suggesting a transition to a band-like regime. For temperatures below 280 K, thermally activated transport is instead observed with $E_a = 61$ meV. Within the thermally activated low temperature range, a reduced $E_a$ is found (from 61 meV to 31 meV) when a low-$\kappa$ dielectric layer like Cytop ($\kappa = 2.1$) is employed instead of PMMA ($\kappa = 3.6$), indicating a dielectric-induced dipolar disorder effect on the DOS, dominating low temperature operation (all activation energy values reported in Supplementary Table 4)[50,51]. Overall, according to a mobility edge model, a scenario is suggested where shallow traps dominate below a threshold temperature of 280 K, above which thermal fluctuations enable charge releasing to extended mobile states[44]. Data on films annealed at $T_1$ and $T_3$ are also displayed in Fig. 7. For both temperatures, purely thermally activated transport is found over the entire temperature range investigated, with extracted activation energies of ≈140 and ≈100 meV for $T_3$ and $T_1$, respectively.

Charge modulation spectroscopy (CMS) and microscopy (CMM) was employed to investigate charge induced optical signatures (i.e. polaronic relaxations) in FETs devices (more details in Supplementary Fig. 32 and Supplementary Note 9). Downstream interpretation of the CMS spectra allows to confidently associate $\Delta T/T < 0$ signals (a narrow one peaking around 1.55 eV and a much broader band with peaks at 0.9 and 1 eV) to polaronic optical transitions and $\Delta T/T > 0$ (between 1.55 and 2 eV) to the bleaching of ground-state absorption (Fig. 8a, b). Negligible electro-absorption effects are present, as detailed in the Supplementary Note 10 and Supplementary Figs. 33 and 34,

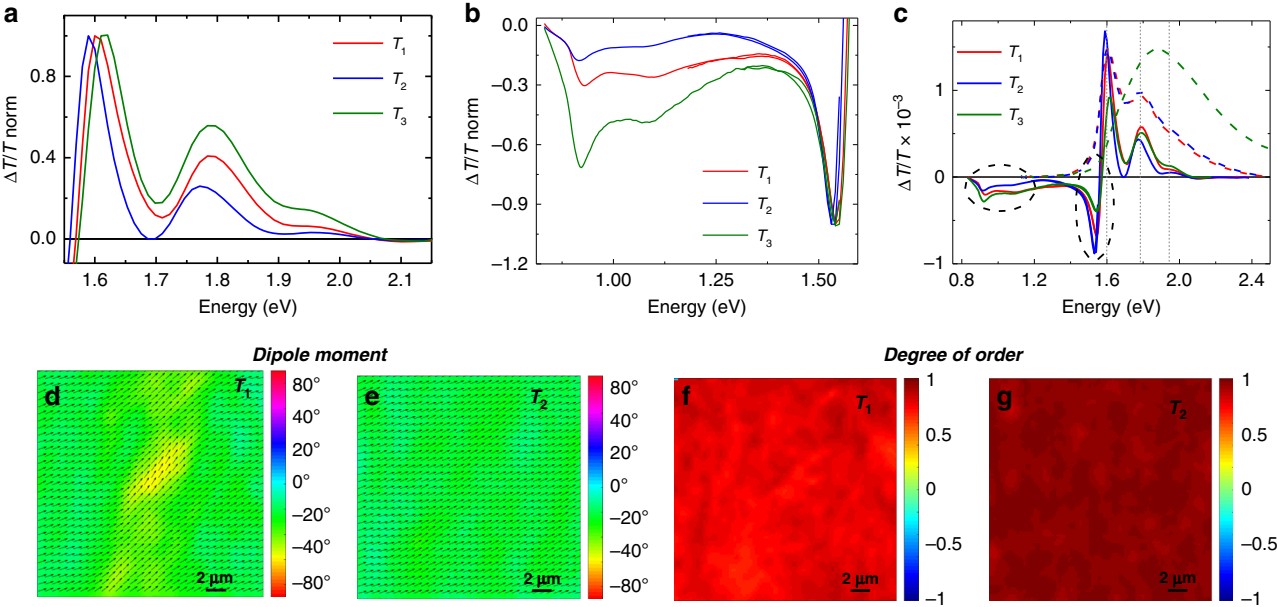

**Fig. 8** Charge modulation micro-spectroscopy. Normalized CMS spectra of PThDPPThF4 in the high energy range (from 1.7 to 2.15 eV) (**a**) and low-energy range (from 0.5 to 1.55 eV) (**b**); **c** CMS spectra of PThDPPThF4 annealed at $T_1$, $T_2$ and $T_3$, driven at $V_{GS} = 20$ V; normalized optical density spectra (dashed lines) are also reported; $20 \times 20 \ \mu m^2$ polarized CMM maps with the indication of the polymers backbone orientation (**d**, **e**) and the relative degree of orientational order (**f**, **g**) of PThDPPThF4 films annealed at $T_1$ (**d**, **f**), and $T_2$ (**e**, **g**)

where an accurate investigation aimed at evaluating the presence of a Stark effect due to electric field modulation within the semiconductor is reported. While the positive peaks at 1.6 and 1.75 eV clearly correspond to the 0–0 and 0–1 vibronic transition bands of UV-vis spectra and thus are related to crystalline regions, the shoulder at 1.9 eV corresponds to the optical response of the non-aggregated phase within the solid films. Interestingly, the 0–0 and 0–1 bands prevail also in films processed at $T_3$, where the bulk optical density is instead totally dominated by the high energy transitions of the amorphous-like phase (green dashed line in Fig. 8c). Thus, even in highly disordered films, charges preferentially select the smaller energy gaps of the residual ordered phase. Nevertheless, it may also indicate some segregation of the residual crystalline phase on the top surface of the film, as also suggested by NEXAFS analysis.

The relative intensities of polaronic bands at 1.55 and ~1.0 eV strongly depend on the annealing temperature and thus on the microstructure. Films annealed at $T_3$, which show the highest structural and conformational disorder and display the strongest thermal activation of transport, exhibit the highest charge absorption around 1.0 eV, comparable to the 1.55 eV absorption. Annealing at $T_2$, where the highest order and the best transport properties are found, leads to the lowest absorption around 1.0 eV, with a strong prevalence of the band at 1.55 eV. The spectral signatures of films annealed at $T_1$ are in between the two previous cases, consistently with mild thermal activation of transport.

The bleaching at 1.9–2.0 eV increases going from efficient transport ($T_2$) to inefficient transport ($T_3$). Indeed, with increasing disorder of the film, the disordered phase is necessarily more involved, leading to larger energy barriers for transport.

By combining CMS with a confocal microscope using a polarized probe and fixing the energy at the bleaching main peak (1.6 eV), it is possible to map the orientation of ground-state transition dipole moments within the channel of FET devices (Fig. 8d–g and Supplementary Fig. 35) and extract the degree of orientational order (DO)[52]. A marked increase in DO is measured in the case of $T_2$ annealed films (DO = 0.98 at $T_2$ vs. DO = 0.89 at

$T_1$), in agreement with the relative improvement of transport properties and with a picture of stronger interconnectivity of the ordered phase in the chain alignment direction upon annealing within the melting endotherm.

**DFT calculations for oligomers and aggregates.** The torsional conformation subspace of single chain PThDPPThF4 was investigated at the DFT level (Supplementary Figs. 36–38 and Supplementary Note 11). Two most stable oligomers that are almost degenerate in energy and characterized by syn- and anti- conformations of the Th-F4-Th units were found. In both, the sulphur atom is on the same side as the lactam-N of the DPP unit. These findings support the structural analysis by solid-state NMR, which provides evidence for the anti- conformation. The latter was therefore adopted to optimize physical (van der Waals) dimers (i.e. aggregates) to calculate the most stable packing structure and inter-molecular interactions. Two dimers were investigated (Supplementary Fig. 38), called H- and J-dimers due to their co-facial or slide packing structure, respectively. In the J-dimer, the DPP unit of one chain interacts with the Th ring of another chain at an intermolecular distance of 3.439 Å. In the H-dimer, the DPP unit of one chain interacts with the DPP of another one at a distance of 3.410 Å. Again in agreement with NMR observations, J-dimer stacking is predicted to be more stable by 5 kcal/mol than the cofacial, H-dimer one.

Aiming at understanding the properties of charged states, we computed the structure and the optical properties of the most stable charged single oligomers and dimers by means of DFT and TD-DFT calculations in the unrestricted scheme (UDFT and TD-UDFT, see Supplementary Figs. 39–41). From structural relaxations and polaron spin density analysis, the polaron localizes over three repeat units on the single chain, regardless the oligomer conformation[53–55]. By considering the most stable charged dimers, H- and J-dimer result in different spin delocalization. In the H-dimer, the spin is localized on a single chain featuring a prevalent intra-molecular character, whereas in the J-dimer the

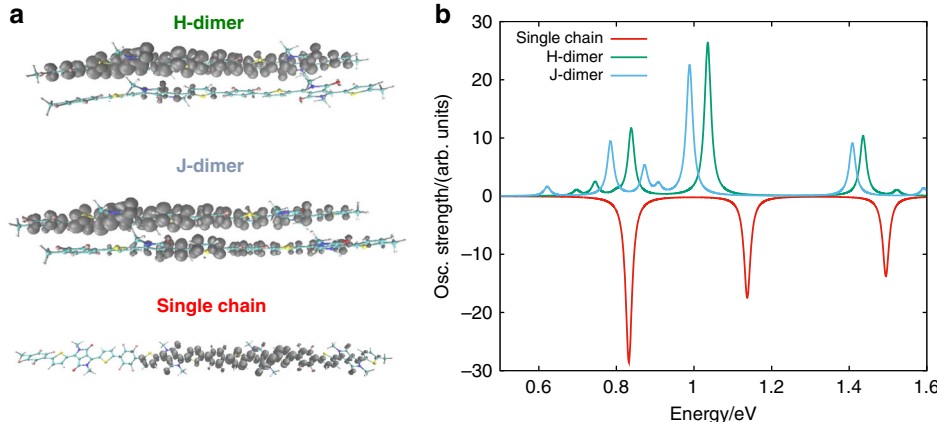

**Fig. 9** Spin density distribution. **a** Computed UDFT (ω-UB97X-D/6-311 G*) spin density for the charged (−1) species of the single chain and H- and J-dimers (both alpha and beta spin densities are highlighted in grey, isosurface values 0.0003 a.u.). **b** Computed TD-UDFT optical polaron transitions for single chain (red), H- (green) and J-dimer (cyano)

polaron spin density is inter-molecularly delocalized, as reported in Fig. 9a. Computed polaron optical transitions at the TD-DFT level are reported in Fig. 9b for the single chain oligomer, H- and J- dimers. Care should be taken regarding band assignments, because TD-DFT calculations of charged conjugated species can drastically underestimate the energy of the electronic transitions (although this effect should be alleviated by using range-separated functionals)[56]. Moreover, the band intensity can also be miscomputed, especially for high energy and/or diffuse (Rydberg character) excited states[57–59]. However, the low-energy transition range can be confidently assigned to and compared with the experimental CMS transitions[60]. Both the single oligomer and dimers show low-energy polaron transitions in the 0.6–0.9 eV region. For the oligomer we computed one intense band at 0.82 eV, mainly related to transitions involving the (highest) SOMOα and the (lowest) SUMOα (i.e., singly occupied/unoccupied molecular orbital). The H-dimer features the same polaron transition, however with a reduced oscillator strength than the oligomer. For the J-dimer, the transition is red-shifted with respect the two previous cases, at 0.79 eV, with a lower oscillator strength as compared to the single oligomer case.

The computed band at 0.8 eV can be assigned to the CMS band observed in the NIR region around 1.0 eV. For $T_3$ processed films, optical behavior is associated with the characteristics of a single chain, therefore the intensity of the 1.0 eV polaron band is higher than that of the more crystalline films. For the case of $T_1$ or $T_2$ annealed films, namely those with an extended degree of order and/or crystallinity, the intensity of the NIR band around 1.0 eV is partially quenched by inter-molecular interactions and spin delocalization, as computed for aggregates (i.e., J-dimer).

The assignment of the CMS band at 1.55 eV is however not straightforward. TD-DFT calculations on charged dimers predict a polaron transition at 1.0–1.1 eV and at 1.18 eV for the single oligomer. These bands, most probably underestimated in energy, are tentatively assigned to the observed band at 1.55 eV. For such computed transitions, the oscillator strength is higher for the dimers than for the oligomer. This observation correlates with the observed behavior for the CMS band at 1.55 eV, which shows a higher intensity for $T_2$ and $T_1$ with respect to $T_3$ annealed films.

Therefore, DFT calculations predict a J-dimer featuring an inter-molecular polaron spin delocalization, with the low-energy optical transition at 0.8 eV showing a lower oscillator strength than the one computed for the single chain. The higher energy polaron optical transition is calculated at 1.0 eV, featuring a higher oscillator strength than the oligomer case. These

observations suggest that the CMS bands are probing polaron species, which, for $T_1$ and $T_2$ annealed films, are inter-molecularly delocalized, recalling similar observations for polycrystalline small-molecule films based on TIPS-pentacene[61].

## Discussion
For a low-molecular weight, high performance electron transporting copolymer, synthesized free of homocoupling defects by a simple direct arylation polycondensation protocol, we have identified key parameters required to induce a transition from thermally activated to temperature independent and charge density independent, single crystal-like electron transport. We have shown that combined uniaxial alignment and rationally selected thermal annealing procedures of films of PThDPPThF4 strongly enhance FET properties, culminating in a thermally independent transport regime in the proximity of 300 K, i.e. close to room temperature. Fast scanning calorimetry probes thermal transitions in thin films and allows the selection of three characteristic annealing temperatures: $T_1$ below, $T_2$ within and $T_3$ above the melting temperature $T_m$. Upon uniaxial alignment through solution processing and mild annealing ($T_1$), films with a relatively high crystalline content, with planar molecular conformation, and a well-defined supramolecularly oriented organization are obtained. Annealing within the melting temperature range and near $T_m$ ($T_2$) results in oriented crystalline lamellae of increased thickness. Melting and subsequent solidification from $T_3$ erases thermal history and chain orientation, and results in a more amorphous film with fewer, smaller crystals with larger average π-stacking owing to increased conformational disorder. A sketch of the structural modification upon the different annealing conditions is proposed in Fig. 10.

CMS measurements indicate that charged states characterized by strongly different energetic relaxation are involved in electron transport. More relaxed states, linked to higher transport barriers, are associated with the non-aggregated, disordered domains, while less relaxed states, linked to lower transport barriers, are associated with the more ordered, crystalline domains. The relative contribution of molecularly ordered and disordered regions to transport is manipulated using different thermal treatments, resulting in different degrees of energetic disorder within FETs and thus in a strong modulation of the thermal barriers to electron transport. A transport improvement with lamellar thickening, obtained upon annealing at $T_2$, is generally expected in semicrystalline polymers[62], but it has been rarely demonstrated[63,64], especially in solution processed films and FET

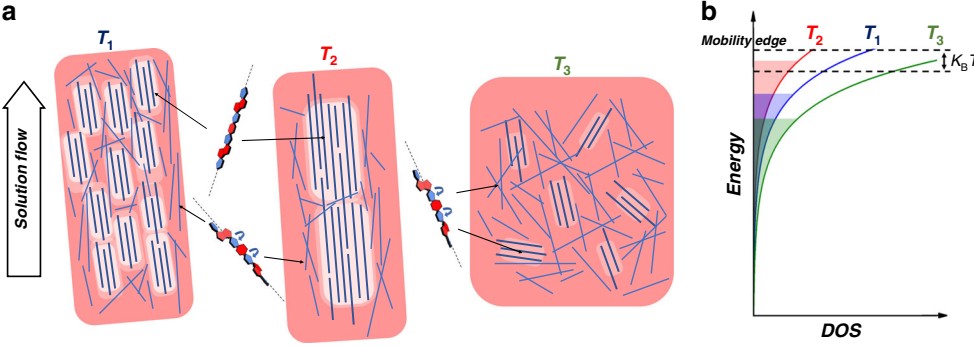

**Fig. 10** Structure-transport property relationship. **a** Proposed sketch for the changes in the functional morphology, i.e. the morphology of the molecules involved in charge transport, of PThDPPThF4 films upon annealing at $T_1$, $T_2$, and $T_3$; thicker lines represent crystalline polymer segments; brighter areas evidence superior electron coupling. **b** Proposed energetic diagram describing transport within films annealed at $T_1$, $T_2$, and $T_3$. According to the mobility edge model, the energy barrier for charge transport is defined as the difference between the mobility edge and the energy of localized states. The DOS tail variation induced by the applied thermal annealing allows for a modulation of transport barriers: thermal annealing at $T_2$ leads to energy barriers inferior to $k_BT$, i.e., around room temperature, thermal fluctuations enable charge releasing from shallow traps to extended mobile states

architectures. Owing to the retention of backbone orientation upon annealing at $T_2$, energetic barriers for charge transfer are strongly reduced exclusively in the direction of chain alignment, where effective interconnectivity of ordered regions can be more easily gained, leading to temperature-independent electron transport near to 300 K. In contrast, in the direction perpendicular to the backbone, thermal charge transport barriers are independent whether the material is annealed at $T_1$ or $T_2$, which is not surprising as in the π-stacking direction coherence length is much smaller in any case. Combining thermal, structural, optical and electrical characterization with computational investigation, a picture can be drawn, in which PThDPPThF4 crystalline domains are energetically highly ordered and allow for superior electron coupling, up to inter-molecular charge delocalization, in virtue of chains co-planarity and, as suggested by ab-initio calculations, J-type aggregation. Moreover, efficient transport through interconnected crystalline domains via chain extended molecules is needed to realize the observed improvement in FET charge transport, as it allows to bypass the high-energy disordered phases present in the film. Insufficient interconnection of crystalline domains limits instead transport in films annealed at $T_1$, and even more drastically in disordered films processed at $T_3$.

These results rationalize the link between complex, semi-crystalline polymer thin film microstructure and more efficient charge transport beyond thermally activated regimes. As such, they can pave the way for a new generation of high performance polymer-based electronics, suitable for a wider range of applications, such as ultra-high resolution displays and wireless communications, currently not accessible because of limited carrier mobilities.

## Methods

**Synthesis**. PThDPPThF4 was synthesized using direct arylation polycondensation involving four steps as described previously. A detailed characterization including molecular weight, main chain defects and end groups is reported elsewhere[26]. The PThDPPThF4 batch used here has SEC molecular weights $M_n/M_w = 14.1/52.2$ kg/mol, is free of homocoupling defects and exhibits mostly hydrogen-termini at either side. Absolute number average degree of polymerization $DP_{n,NMR}$ and number average molecular weight $M_{n,NMR}$ are obtained from the analysis of end groups in $^1H$ NMR spectra, and were estimated to be 8 and 8.2 kg/mol, respectively.

**Fast scanning calorimetry**. The thermal behavior of PThDPPThF4 thin films were studied by fast scanning calorimetry (FSC) (FSC, Flash DSC 1, Mettler Toledo). The samples were prepared by spin coating PThDPPThF4 solutions (5 mg/mL in toluene) at 1000 rpm directly onto the chip sensor used for FSC experiments. Two types of experiments were conducted: (i) the calorimetric signals of the spin coated thin films between 30 and 450 °C were probed at 2000 °C/s by

analyzing the first heating, the second heating and the cooling scans; (ii) annealing experiments were performed following the thermal protocol depicted in Fig. 1c in the manuscript: i.e., the freshly spin coated samples were taken to the suitable annealing temperatures at 2000 °C/s (stage I) and kept at that temperature for 5 min (stage II). Then, the thermal signal was recorded during a subsequent heating scan at 2000 °C/s from the annealing temperatures up to 450 °C (stage III).

**Film topography and thickness**. The surface topography of the films was measured with an Agilent 5500 Atomic Force Microscope operated in the Acoustic Mode. The thicknesses of the polymer films were measured with a KLA Tencor Alpha-Step Surface Profiler. PThDPPThF4 films for topography and thickness analysis where prepared either through off-center spin coating (30 s, 1000 rpm) or through wired-bar coating on silicon p-doped substrates. Uniaxially aligned films have been subjected to thermal annealing at different temperatures (in the 120–350 °C range) for 30 min.

**TEM sample preparation and characterization**. The polymer films were first coated with an amorphous carbon film of a few nm thickness under high vacuum (Edwards Auto 306 evaporator). Then, the carbon-coated films were lifted from the substrate using an aqueous solution of HF (5% in weight) and recovered on TEM copper grids. The samples were analyzed with a FEI CM12 microscope (120 kV) equipped with a Megaview III camera under low dose conditions in bright field and diffraction modes.

**FET fabrication and electrical characterization**. Thoroughly cleaned 1737F glass was used as substrate for all the devices. FETs were fabricated using a top-gate, bottom-contact architecture. Bottom Au contacts were prepared by a lift-off photolithographic process with a 1.5 nm thick Cr adhesion layer. The thickness of the Au contacts was 15 nm. Patterned substrates were cleaned in an ultrasonic bath in isopropyl alcohol for 2–3 min before deposition of the semiconductor. PThDPPThF4 thin films (30–50 nm) were deposited either through off-center spin coating (30 s, 1000 rpm) or through wired-bar coating. Films have been subjected to thermal annealing at different temperatures (in the 120–350 °C range) for 30 min. As the dielectric layer, we have employed either the perfluorinated polymer CYTOP CTL-809 M dielectric (Asahi Glass, spun at 6000 rpm for 90 s), or PMMA (spun at 2000 rpm for 90 seconds); in both cases dielectric film thickness was ∼550 nm. As the gate electrode, thermally evaporated 40 nm thick Al layer or 4.5 nm thick Au transparent layer were employed for standard FETs and CMS/CMM measurements, respectively. A charge injection interlayer was introduced in some devices, by spin-coating a 0.5 g/l P(NDI2ODT2) solution in toluene on top of the source and drain electrodes (1000 rpm, 30 sec) resulting in 2.5 nm ultra-thin films. The interlayer was annealed at 310 °C for 1 h in order to make it insoluble during the successive deposition of PThDPPThF4 from toluene. The electrical characteristics of transistors were measured in a nitrogen glovebox on a Wentworth Laboratories probe station with an Agilent B1500A semiconductor device analyzer. Saturation and linear charge carrier mobility values were extracted by the transfer characteristic curves according to the gradual channel approximation, following the expressions $I_{DS} = \mu_{sat} \times C_{die} \times W/2 L \times (V_{GS} - V_{Th})^2$ and $I_{DS} = \mu_{lin} \times C_{die} \times W/L \times [(V_{GS} - V_{Th}) \times V_{DS} - V_{DS}^2/2]$, where $I_{DS}$ is the drain current, $\mu_{sat}$ and $\mu_{lin}$ are the saturation and linear mobility, respectively, $C_{die}$ is the specific dielectric capacitance, $W$ and $L$ are the width and the length of the channel, respectively, $V_{GS}$ is the gate voltage, $V_{DS}$ is the drain voltage, and $V_{Th}$ is the threshold voltage. Accordingly, $\mu_{sat}$ ($\mu_{lin}$) was obtained from the slope of $I_{DS}^{0.5}$ ($I_{DS}$) versus $V_{GS}$, calculated every three points around each $V_{GS}$ value. The effective mobility $\mu_{eff}$ was obtained by

multiplying μ (alternatively $\mu_{sat}$ and $\mu_{lin}$) by the measurement reliability factor $r$: $\mu_{eff} = r \times \mu$. $r$ was determined according to the definition and equation reported in the work by Choi et al.[45].

**UV-Vis absorption**. The optical absorption measurements were performed using a spectrophotometer Perkin Elmer Lambda 1050. PThDPPThF4 films for UV-Vis where prepared either through off-center spin coating (30 s, 1000 rpm) or through wired-bar coating on glass. Uniaxially aligned films have been subjected to thermal annealing at different temperature (in the 120–350 °C range) for 30 min.

**GIWAXS measurements**. GIWAXS measurements were performed on the SAXS/WAXS Beamline at the Australian Synchrotron[65]. In all, 11 or 18 keV highly collimated photons were aligned parallel to the sample by using a photodiode, with scattering patterns collected on a Dectris Pilatus 1 M detector. Angular steps of 0.01° were taken near the critical angle, which was determined as the angle of maximum scattered intensity. Each 2D scatter pattern was a result of a total of 3 s of exposure. Three 1-s exposures were taken at different detector positions to fill in the gaps between modules in the detector, and combined in software. Correction of data onto momentum transfer axes and sector profiles, was done using a modified version of NIKA. Peak fitting was done using least squares multipeak fitting within Igor Pro. By identifying the scattering peaks, and fitting them with a log cubic empirical background function, we can monitor the crystallinity in the three major unit cell directions. Peak area is proportional to the number of molecular planes which participate in crystalline stacking and so the relative crystallinity within the thin film, while the peak width is an indication of the coherence length of that crystallinity and represents the size and quality of those crystals, and finally the location of the peak gives the d-spacing or the distance from one unit cell to the next along that direction.

PThDPPThF4 films for GIWAXS where prepared either through off-center spin coating (30 s, 1000 rpm) or through wired-bar coating on silicon p-doped substrates. Uniaxially aligned films have been subjected to thermal annealing at different temperature (in the 120–350°C range) for 30 min.

**NEXAFS measurements**. NEXAFS measurements were performed on the Soft X-ray Beamline at the Australian Synchrotron[66]. Partial electron yield (PEY) was used to collect the data, with the PEY signal measured with a channeltron detector with a retarding voltage of ~200 V. Dark levels were measured and subtracted from the data, and double normalization was done using a gold mesh corrected by a photodiode. Peak fitting and tilt angle calculations were done using the Quick AS NEXAFS Tool[67].

All tilt angles are calculated from a peak fit of structure at lower energies than 286 eV, which is in the π* manifold, corresponding to transition dipole moments normal to the face of a conjugated core, along the direction of the carbon π orbitals. Thus a tilt angle of 90° indicates that every TDM is perfectly in-plane, and so all the conjugated faces are oriented perfectly edge-on to the substrate, while a tilt angle of 0° indicates perfectly face-on orientation. Because of the uniaxial alignment of spin coated films, an unaligned film will have an average tilt angle of ~54.7°. It is also impossible to determine through only a tilt angle measurement the distribution of local tilt angles within the film, but only the ensemble tilt angle. Thus any fixed dihedral angle between backbone components will result in an average tilt angle between the tilt angles of each component. As opposed to GIWAXS, NEXAFS is sensitive to all molecules, regardless of them being in a crystalline or amorphous region. PThDPPThF4 films for NEXAFS were prepared either through off-center spin coating (30 s, 1000 rpm) or through wired-bar coating on silicon p-doped substrates. Uniaxially aligned films have been subjected to thermal annealing at different temperature (in the 120–350 °C range) for 30 min.

**Charge modulation spectroscopy**. The CMS spectra were collected in working FETs, measuring the normalized transmittance variation ($\Delta T/T$) induced by the gate voltage modulation. We performed the measurements keeping the source and drain electrode at 0 V, while the modulated voltage ($f = 983$ Hz) was applied at the gate electrode. The offset voltage and amplitude of modulation varied depending on the sample. Details on these values are found in figure captions. The light of a tungsten lamp was monochromated and subsequently focused on the device. The transmitted portion of the light was then collected and revealed through a silicon photodiode. The electrical signal is amplified through a trans-impedance amplifier (Femto DHPCA-100) and then revealed through a DSP Lock-in amplifier (Standford Instrument SR830). All the measurements were performed in vacuum atmosphere (~$10^{-5}$–$10^{-6}$ mbar).

**Polarized charge modulation microscopy**. The p-CMM data were collected with a homemade confocal microscope operating in transmission mode. The light source consisted of a supercontinuum laser (NKT Photonics, SuperK Extreme) monochromated by an acousto-optic modulator (NKT Photonics, SuperK Select) in the 500–1000 nm region with line widths between 2 and 5 nm. Laser polarization was controlled with a half-wave plate and a linear polarizer. The light was then focused on the sample with a 0.7 N.A. objective (S Plan Fluor60, Nikon) and collected by a second 0.75 N.A. objective (CFI Plan Apochromat VC 20, Nikon). The collected light was focused to the entrance of a multimodal glass fiber with a 50 μm core, acting as a confocal aperture. Detection was operated through a silicon

photodetector (FDS100, Thorlabs). The signal was amplified by a transimpedance amplifier (DHPCA-100, Femto) and supplied both to a DAQ (to record the transmission signal, $T$) and to a lock-in amplifier (SR830 DSP, Stanford Research Systems) to retrieve the differential transmission data, $\Delta T$. The system was run with an homemade Labview software. The OD values were obtained from $T$ as described in ref. [52], while the CMS data were calculated as $\Delta T/T$. The sample was kept in an inert atmosphere by fluxing nitrogen in a homemade chamber. The gate voltage of the transistor was sinusoidally modulated at 989 Hz between 20 and 60 V with a waveform generator (Keithley 3390) amplified by a high-voltage amplifier (Falco Systems WMA-300), while the source and drain contacts were kept at short-circuit. Signal maps were collected by raster scanning the sample at 250 ms per pixel, with a lock-in integration time of 100 ms. All data were processed with Matlab software.

**Solid-state NMR**. All $^1$H and $^{19}$F solid-state NMR experiments were performed on a Bruker DSX 500 spectrometer operating at a magnetic field of 11.74 T, corresponding to Larmor frequencies of 500.2 MHz ($^1$H) and 470.6 MHz ($^{19}$F). All samples were packed into 2.5 mm o. d. $ZrO_2$ rotors with Vespel drive and bottom caps. The $^1$H and $^{19}$F Magic Angle Spinning (MAS) NMR experiments employed a commercial Bruker 2.5 mm MAS triple resonance probe and a Bruker 2.5 mm MAS double resonance probe with low fluorine background, respectively, using MAS frequencies of either 25.0 or 29.762 kHz. The $^1$H and $^{19}$F chemical shift and corresponding radio-frequency fields were calibrated using adamantane and PTFE (−122 ppm). A π/2 pulse length of 2.5 μs ($^1$H) and 3 μs ($^{19}$F) was used for excitation, corresponding to RF field strengths of 100 and 83.3 kHz, respectively. The magic angle was calibrated using KBr.

The Back-to-Back (BaBa) DQ excitation and reconversion scheme was used in combination with a XY-8 phase cycle[68] for both the 2D $^1$H-$^1$H and $^{19}$F-$^{19}$F DQ-SQ correlation NMR experiments. Both the 2D $^1$H-$^1$H and $^{19}$F-$^{19}$F DQ-SQ experiments employed four rotor periods of DQ excitation and reconversion and the indirect dimension was incremented in a rotor-synchronized fashion; phase discrimination was achieved via the STATES-TPPM scheme[69] and 64 $t_1$-increments were acquired in the indirect dimension. A short z-filter delay of one rotor period was applied prior to acquisition. Acquisition, processing, and data analysis were performed using the Bruker Topspin software package. All 2D DQ-SQ-correlation spectra were apodized with a shifted Gaussian window function (GB = 0.25; LB = -100 Hz ($^1$H) and −500 Hz ($^{19}$F)) to improve resolution in both dimensions.

**Computational methods**. DFT calculations of neutral and charged (−1) species were performed using the range-separated functional ω-B97X with D2 Grimme dispersion corrections (ω-B97X-D) and the 6-311 G* basis set. TDDFT calculations were performed at the same level of theory considering at least 30 excited states for each species. The unrestricted U-DFT formalism was adopted for the charged states. We investigated the oligomer lengths $n = 1$, 2 and 4. Molecular dimers were made starting from the oligomer $n = 2$. All structures (i.e., oligomers and dimers) were optimized and the evaluation of the vibrational frequencies (i.e. Hessian analysis) were performed to assure the stable equilibrium conformation. All Cartesian coordinates of the optimized structures are available under request to the co-author D.F. All calculations were performed with Gaussian09/D.01 code[70].

## Data availability
The authors declare that the main data supporting the findings of this study are available within the article and its Supplementary Information files. Extra data are available from the corresponding authors upon request.

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

## Acknowledgements

M.C. thanks D. Natali for insightful discussions. M.S. and F.N. thank M. Hagios and A. Warmbold for SEC and bulk DSC measurements, respectively. This work was financially supported by the European Research Council (ERC) under the European Union's Horizon 2020 research and innovation programme "HEROIC", grant agreement 638059. M.S. acknowledges funding from the DFG (SO 1213/8-1). N.S. acknowledges the financial support of the US National Science Foundation through the DMREF program (DMR-1729737). D.F. acknowledges the Deutsche Forschungsgemeinschaft (DFG) for a Principal Investigator grant (FA 1502/1-1). Part of this work was carried out on the SAXS/WAXS and Soft X-ray Beamlines at the Australian Synchrotron, part of ANSTO and at Polifab, the micro- and nano-technology center of the Politecnico di Milano.

## Author contributions

A.L., M.S. and M.C. conceived and coordinated the work. A.L. deposited the films, fabricated the field-effect devices, and took care of the whole electrical characterization. A.L. performed CMS and CMM experiments. Electrical, CMS, and CMM data were analysed by A.L. and M.C. F.N. and M.S. contributed the polymers under study. E.G. and C.R.M. performed X-Ray measurements and analysed structural data. J.M. and N.S. took care of thermal and POM characterizations, and analysed the resulting data. D.F. performed DFT calculations. P.S. and M.R.H. performed the solid-state NMR study. M.B. performed TEM measurements and contributed to the analysis of the structural data.

A.L., J.M., D.F., P.S., C.R.M., M.R.H., M.S. and M.C. co-wrote the paper. All authors contributed to the editing of the paper.

## Additional information

**Competing interests:** The authors declare no competing interests.

