## [Peer Review File · Nature Communications]

Reviewers' comments:

Reviewer #1 (Remarks to the Author):

In this work, the charge carrier transport of electron transporting diketopyrrolopyrrole tetrafluorobenzene copolymer was investigated in terms of its film morphology. The morphology was controlled by alignment and annealing. It is claimed that annealing in the melting endotherm of the aligned films leads to voltage-independent and non-thermally activated electron transport. Based on the experimental results, the authors used computational methods to conclude the transport mechanism.

This is a comprehensive work providing good insight on the correlation between charge transport and polymer morphology since the authors applied a broad range of methods for the analysis.

However, before reconsideration for publication the following points need revision:

- Figure 7 contains only 6 data points. But already these results show that for the T2 sample the mobility decreases upon lowering the temperature. Therefore, the title of the manuscript is misleading (... temperature independent electron transport ...). Typically, the band (or band-like) transport observed for organic semiconductors (like in e.g. <https://doi.org/10.1073/pnas.1705164114>) shows reverse mobility / temperature relation (or at least the mobility remains unchanged) within a much broader temperature range (below 220 K). Despite a decrease of the activation energy and increased transport efficiency for the T2 samples, there is no sufficient evidence for a band (or band-like) transport. It would be also important to add error bars for these measurements.
- What is the role of the face-on crystallites on the charge transport in T2 samples? It could be assumed that this crystallite population is mainly responsible for the improved charge transport due to 3D conduction (see: J. Am. Chem. Soc., 2011, 133, 20130–20133)? This point needs a deeper analysis since the XRD study reveals only a small increase in crystallite size for T2 in comparison to T1. The coherence length increases only by one unit from 13.8 nm ($13.8/2.04=6.8$ units) to 16.5 nm ($16.5/2.14=7.7$ units).
- Authors refer to hopping-transport without specifying the model they refer to. Do authors have gaussian-disorder model in mind? This should be specified since in Figure 11 the authors mention the mobility-edge model which differs from the mentioned GDM model.
- The authors ascribe the improvement of charge transport to enhanced crystalline order of T2 samples. However, recent studies on relation between the microstructure and charge transport in conjugated polymers, proves that the most critical factor for enhanced charge transport is level of lattice disorder in the pi-stacking direction within the crystallites, rather than amount of crystalline phase within the sample and size of the crystalline domains (DOI: 10.1038/NMAT3722). Is it possible that authors could also estimate or measure lattice disorder level?

Reviewer #2 (Remarks to the Author):

Conventionally, charge transport in semi-crystalline donor-acceptor conjugated polymers is thermally limited – the corresponding electrical performance is dependent on the effectiveness of interchain hopping. In this manuscript, the authors reported that non-thermally activated, single crystal-like electron transport could be realised with the assistance of uniaxial alignment and carefully controlled thermal annealing. High performance is generally envisaged using alignment processing and post annealing at proper temperature, however, findings of non-hopping limited transport are rare, even under optimised device conditions. Though gate voltage independent mobility of up to $\sim 3 \text{ cm}^2\text{V}^{-1}\text{s}^{-1}$ with reliability factor of up to $\sim 80\%$ does not seem striking amongst these high mobilities nowadays, deep understanding of microstructure related charge transport property is provided in this manuscript. Findings in this paper may provoke the designing of new high performance polymers for organic transistors. I therefore recommend acceptance after revision.

I have the following questions to be addressed by the authors.

1. The off current of FETs in Figure 6b are at relatively high level, this happens quite often for

ambipolar transistors. The off current also obviously depends on annealing temperature. Is the higher channel current associated with higher gate leakage and thus resulted in higher off level?

2. The V_{on} after annealing at higher temperature (T2 and T3) shifted to the positive bias direction (Figure 6b). Is it attributed to delamination at the semiconductor/electrode interface? Did V_{on} also shift from as-spun to T1 annealed devices?
3. The authors mentioned using P(NDI2ODT2) as interlayer to improve injection of electrons. P(NDI2ODT2) is however a semiconducting layer itself, which makes determination of its contribution to the overall performance of transistors difficult. I would suggest a nonconductive injection layer instead to see the possibility of further optimisation.
4. Page 13 line 379 and 380, the number of activation energy should be 61 meV instead of 66 meV according to Figure S26 and Table S4.

Reviewer #3 (Remarks to the Author):

Alessandro Luzio et al. provided an interesting topic on "Microstructural control enables temperature independent electron transport in polymer transistors". Some ideas are very meaningful such as residual energy barriers can be bypassed in the direction of molecular alignment. Chain alignment and crystal thickness can be controlled by annealing temperature before melting point. However, they provided limited data for their conclusions.

1. Although authors provide the relationship between mobility and temperature in figure 7, it is really difficult to convince the band-like transporting in their polymers. Firstly, the value of mobility is stable during the changes of $1000/T$ from 3.35 (298.5K) to 3.6 (277.8K). Since the change of T is so small, it is difficult to know the trend of mobility changes. Normally, for band-like transporting behavior, there should be a max mobility peak in their curve. So if authors want to confirm the transporting mechanism, they should provide mobility values in a range of 100 K changes. Such as in *Adv. Mater.* 2016, 28, 5276–5283 from 100 K to 350 K.

2. So, it is also difficult to say "temperature independent electron transport in polymer transistors" based on present data.

3. Is it better to limit the types of polymers? What is the universality of this annealing process? Is it also applicable to classical NDI based D–A materials, such as N2200? Too big a topic can easily mislead the reader.

4. Since molecular weight is very important for polymer properties (PThDPPTThF4, M_n : 14 kg mol⁻¹; dispersity, \bar{D} : 3.7). Please provide the GPC data for PThDPPTThF4. How does the molecular weight effect on their FET mobility? The detail molecular structure is lacking for PThDPPTThF4, such as what is kind of R group?

Some details for FET fabrication are missing, such as output curves of their devices; calculating equation of effective mobility; how to prepare an interlayer of P(NDI2ODT2) on electrode (Figure S25), does it not destroy during PThDPPTThF4 coating? In main text, figure S26 is for PThDPPTThF4 or N2200?

General remarks

We take a chance in the first place to thank the reviewers for taking time to carefully revise our work and to raise good points of discussions. We addressed all their questions with care and, where applicable, modified the manuscript as described in detail here below.

We believe this review process was very helpful to improve our work and to clarify a few major points.

Reviewer #1

In this work, the charge carrier transport of electron transporting diketopyrrolopyrrole tetrafluorobenzene copolymer was investigated in terms of its film morphology. The morphology was controlled by alignment and annealing. It is claimed that annealing in the melting endotherm of the aligned films leads to voltage-independent and non-thermally activated electron transport. Based on the experimental results, the authors used computational methods to conclude the transport mechanism.

This is a comprehensive work providing good insight on the correlation between charge transport and polymer morphology since the authors applied a broad range of methods for the analysis.

We thank the reviewer for the positive consideration of our work.

However, before reconsideration for publication the following points need revision:

- Figure 7 contains only 6 data points. But already these results show that for the T2 sample the mobility decreases upon lowering the temperature. Therefore, the title of the manuscript is misleading (... temperature independent electron transport ...). Typically, the band (or band-like) transport observed for organic semiconductors (like in e.g. <https://doi.org/10.1073/pnas.1705164114>) shows reverse mobility / temperature relation (or at least the mobility remains unchanged) within a much broader temperature range (below 220 K). Despite a decrease of the activation energy and increased transport efficiency for the T2 samples, there is no sufficient evidence for a band (or band-like) transport. It would be also important to add error bars for these measurements.

Thank you for the comment. This aspect is central to our work and we therefore aim at clarifying these concerns fully.

First, we would like to clarify the difference between a so called “band-like” and a temperature independent transport regime. The reviewer is perfectly correct in stating that in a “band-like” regime an inverse temperature dependence is observed. This is the case of the suggested paper, an interesting work by Jorchescu et al. that refers to p-type devices based on a small-molecule semiconductor, which we have now cited in our manuscript (page 2).

However, an inverse temperature dependence is not the case for our devices, and therefore we are not claiming to observe “band-like” transport. Instead, what we observe is that for specific processing conditions electron mobility becomes independent from temperature when the temperature is high enough, i.e. close to room temperature. Our claim is therefore that, close to room temperature, we are able to generate temperature independent transport in our field-effect devices. We interpret this as a transition regime from purely temperature activated to band-like transport, which may occur at higher temperatures. Higher temperatures are not accessible, since a temperature well above room temperature may induce morphological changes. On the other hand, below a critical temperature that depends on the energetics of the specific semiconductor, electron transport is again thermally activated. These two boundaries are restricting the observed regime within which mobility is constant.

It is not possible to add error bars in a meaningful way in Figure 7, as each data set requires very long measurements. We have instead indicated in Table S4 the error in the fitting when extracting the activation

energy, which does not result in higher values than a few meV. This already indicates that we can distinguish very small activation energies. Indeed, it is simple to distinguish temperature activated transport, even with low activation energy (as also shown in Figure 7), from temperature independent transport. It is evident from Figure 7 that only in the case of annealing at T_2 mobility is constant for temperatures between 300 K to 280 K. Even a small activation energy of 31 meV, as found for temperatures below 270 K, would have produced a clearly visible slope. It is important to note that in Figure 7 we are reporting that mobility is constant from 300 K to 280 K for two different devices with two different dielectrics (CYTOP and PMMA).

Moreover, to further support and confirm our claim, we have gathered more data on devices with aligned films annealed at T_2 . In Figure R1 below, we report two new sets of variable temperature transfer curves measured on FETs with PMMA (panel a and c). In one of the two sets we extend the temperature range to 310 K, adding one point with respect to the range in the paper reaching 300 K (panel a). As commented above, moving to higher temperatures is not viable, as hysteretic behavior starts to appear. Moreover, in Figure R1 we report also the same CYTOP data of the paper, where we added a previously acquired further point at 310 K (panel b). In panel c, the mobility values vs. temperature are shown, confirming what we have previously reported.

Therefore, the temperature independence of electron mobility in a range close to room temperature for FET based on aligned films annealed at T_2 is robust and reproducible.

We have added the data reported in Figure R1 in the Supporting Information (SI) as Fig. S31, and have referred to it in the manuscript, on page 13.

Having clarified our claim and further supported its solidity, we deem it important to avoid possible misunderstandings. Therefore, we have modified the title of the manuscript, specifying that temperature independent transport achieved by the suppression of thermal barriers, is observed close to room temperature. The new title is:

“Microstructural control suppresses thermal activation of electron transport at room temperature in polymer transistors”

We have also carefully revised expressions used throughout the main text. We note that the observation of temperature independent transport close to room temperature is clearly underlined in several points, e.g. as in the abstract and in the conclusions.

Nevertheless, we amended the text for clarity in several parts. In particular, on page 2, we modified the following paragraph:

“We demonstrate that the combination of uniaxial alignment and thermal annealing within the temperature range where the melting of crystals occurs gives access to efficient electron transport at the boundary between hopping, i.e. temperature-activated, and temperature-independent, i.e. band-like regimes.”

By rephrasing:

“We demonstrate that the combination of uniaxial alignment and thermal annealing within the temperature range where partial melting of crystals occurs gives access to efficient temperature-independent electron transport close to room temperature, at the boundary between temperature-activated and band-like regimes.”

Figure R1. (a-c) Square root of the saturation ($V_D = 60$ V) transfer curves at variable temperature, where the solid lines are the measured data and the dashed lines are linear fits according to the gradual channel approximation. d) μ vs. $1000/T$ plot of T_2 annealed and aligned PThDPPTThF4 films with different channel lengths and dielectric layers.

- *What is the role of the face-on crystallites on the charge transport in T_2 samples? It could be assumed that this crystallite population is mainly responsible for the improved charge transport due to 3D conduction (see: *J. Am. Chem. Soc.*, 2011, 133, 20130–20133)? This point needs a deeper analysis since the XRD study reveals only a small increase in crystallite size for T_2 in comparison to T_1 . The coherence length increases only by one unit from 13.8 nm ($13.8/2.04=6.8$ units) to 16.5 nm ($16.5/2.14=7.7$ units).*

We understand the point of discussion raised by the reviewer. In the cited JACS paper a scenario of 3D conduction is speculated for a system where there is mixed face-on and edge-on packing. However, this is a completely open debate, as for example other works directly investigating charge density distribution in polymer FETs claim a strict 2D scenario (e.g. *Adv. Mater.* 26, 728-733, 2014). As a matter of fact, we are very far from a consensus on this point.

A deeper analysis of our data based on the models proposed in such literature is also not possible, because such models are derived under the assumption of thermally activated mobility. With annealing at T_2 we reveal a different regime, for which quantitative models have yet to be developed.

The reviewer is correct in stating that the coherence length increases by a single unit in the lamellar stacking direction. While in this crystallographic direction this modification appears limited, the same is accompanied by a bigger increase in the π -stacking direction, where the increase is of roughly 5 units, confirming an enlarged dimensions of the crystalline regions. Therefore, overall we have evidence of a genuine microstructural evolution upon annealing at T_2 , in which increase of coherence length is one of the effects, together with the packing motif rearrangement.

- *Authors refer to hopping-transport without specifying the model they refer to. Do authors have gaussian-disorder model in mind? This should be specified since in Figure 11 the authors mention the mobility-edge model which differs from the mentioned GDM model.*

We thank the reviewer for this comment. The regime in which mobility is thermally activated up to room temperature, as in the case of T_1 and T_3 annealed films in our work, is common to most polymer field-effect transistors reported to date. Several models have been proposed to explain such behavior, including different hopping mechanisms and mobility edge. This is a still open debate, and it is not the purpose of our work to settle such debate. Our goal was to show how to achieve a different transport regime, where mobility is no longer thermally activated close to room temperature.

We'd like also to note that using an Arrhenius dependence of mobility with temperature is not sufficient to discern between a hopping or a mobility edge model, as both can provide good fitting, yet with very different physical basis (see for example Bredas et al. *PHYSICAL REVIEW B* 87, 195209, 2013).

The reviewer in any case correctly highlights that we have often used the term "hopping" to describe the thermally activated regime, while our proposed mechanism, as depicted in Figure 11 and clearly indicated already starting from page 13, is based on a mobility edge model. This created confusion and we have carefully revised the text by replacing "hopping" with "thermally activated transport" for the sake of generality.

After this clarification, it is worth to discuss why we chose in the end a mobility edge picture to describe what we have observed. First, according to some of the most recent literature in the field, charge transport in high-mobility transistors based on semicrystalline as well as amorphous-like donor-acceptor copolymers

films is described within a mobility edge framework. This is the case for example in Noriega et al. paper (*A general relationship between disorder, aggregation and charge transport in conjugated polymers. Nat. Mater.* 12, 1038, 2013), where the mobility edge model is adopted to describe thermally activated transport. Second, in our work we have evidence of thermally independent transport close to room temperature for T_2 annealed films, pointing to the presence of more delocalized states above a tail of localized states. This is not compatible at all with a hopping picture, where transport occurs only through thermal activated tunneling in between localized states. As a conclusion, in order to provide a plausible picture to describe our observation, without any attempt to settle such complicated and open debate, we opted for the most consistent option considering the current knowledge.

- The authors ascribe the improvement of charge transport to enhanced crystalline order of T2 samples. However, recent studies on relation between the microstructure and charge transport in conjugated polymers, proves that the most critical factor for enhanced charge transport is level of lattice disorder in the pi-stacking direction within the crystallites, rather than amount of crystalline phase within the sample and size of the crystalline domains (DOI: 10.1038/NMAT3722). Is it possible that authors could also estimate or measure lattice disorder level?

This is an important point. One key aspect to take into account when considering the mentioned paper, is that such regime, in which paracrystallinity rules transport, is established only when sufficient interconnectivity among crystalline domains is present. Such interconnectivity is linked to polymer molecular weight. In the same paper it is shown that a high molecular weight (Figure 4 in the paper by Salleo and coworkers) is required to enter such regime, thanks to the presence of tie chains.

In our work we have used a rather low molecular weight polymer ($M_{n,NMR} = 14 \text{ kg mol}^{-1}$). For such a low molecular weight of a polymer with rigid backbone it is likely that extended chain crystals are present and tie chains absent. Such scenario therefore largely deviates from the one where paracrystallinity dominates.

As highlighted in the conclusion, our understanding is that in this scenario the less crystalline interphase between crystalline domains limits transport for films annealed at T_1 , as an effect of limited interconnectivity and the absence of tie chains. Only through annealing at T_2 of aligned films and the consequent growth of the crystalline phase, better interconnectivity in the alignment direction is achieved (see Figure 11). Thus, the more disordered phase can be bypassed and transport is optimized. Very importantly, the alignment direction in which transport improves is perpendicular to the π -stacking direction.

As a further difference from the cases described in the referenced paper, we note that in the work by Noriega et al., even if high-mobility polymers with high molecular weight are described, only examples of thermally activated transport are reported. In our case instead, we show how to achieve temperature independent transport close to room temperature with a low molecular weight polymer.

Directly answering to the final question, the characterization of lattice disorder, i.e. the estimation of paracrystallinity, cannot be unfortunately obtained with the available data. This is due to the fact that we observe only one π -stacking peak, while such analysis is typically performed on multiple orders reflections. While the method proposed by Warren and Averbach (Phys. Rev. B 84, 045203, 2011 and papers cited therein) allows such analysis thanks only to the Fourier transform of the π -stacking peak, strict conditions apply to the data acquisition and characterization of the scattering background is required, making it the method not applicable in our case.

Reviewer #2

Conventionally, charge transport in semi-crystalline donor-acceptor conjugated polymers is thermally limited – the corresponding electrical performance is dependent on the effectiveness of interchain hopping. In this manuscript, the authors reported that non-thermally activated, single crystal-like electron transport

could be realised with the assistance of uniaxial alignment and carefully controlled thermal annealing. High performance is generally envisaged using alignment processing and post annealing at proper temperature, however, findings of non-hopping limited transport are rare, even under optimised device conditions. Though gate voltage independent mobility of up to $\sim 3 \text{ cm}^2\text{V}^{-1}\text{s}^{-1}$ with reliability factor of up to $\sim 80\%$ does not seem striking amongst these high mobilities nowadays, deep understanding of microstructure related charge transport property is provided in this manuscript. Findings in this paper may provoke the designing of new high performance polymers for organic transistors. I therefore recommend acceptance after revision.

We thank the reviewer for the positive feedback on our work.

I have the following questions to be addressed by the authors.

1. The off current of FETs in Figure 6b are at relatively high level, this happens quite often for ambipolar transistors. The off current also obviously depends on annealing temperature. Is the higher channel current associated with higher gate leakage and thus resulted in higher off level?

The off current in the reported devices is not due to gate leakage but to ambipolarity. As an example, we compare in the graph below the transfer (blue) and gate leakage (red) curves of a FET based on a T_2 annealed PThDPPTThF4 film.

Figure R2. Typical transfer characteristic curve of an optimized transistor in saturation ($V_D = 60 \text{ V}$). In blue the channel current, and in red the gate leakage current. The “off” current below 10 V is evidently a p-type branch and not related to gate leakage.

We have clarified the source of the off-current in the manuscript on page 12, and have added examples of transfer curves including gate leakage in Supporting Information as Figure S25 and S26 (Section 2.10) to provide a representative example for gate leakage.

2. The V_{on} after annealing at higher temperature (T_2 and T_3) shifted to the positive bias direction (Figure 6b). Is it attributed to delamination at the semiconductor/electrode interface?

Thanks for the comment. We cannot exclude such possibility, however we are not able to prove it as we did not perform specific experiments. We note that the very same microstructural evolution that we observe could produce a voltage shift as it affects the contact region both at the interface and in the access bulk region.

Did Von also shift from as-spun to T1 annealed devices?

We do not have such evidence since to obtain consistent and reproducible results we always applied a 120 °C annealing throughout the whole study.

3. The authors mentioned using P(NDI2ODT2) as interlayer to improve injection of electrons.

P(NDI2ODT2) is however a semiconducting layer itself, which makes determination of its contribution to the overall performance of transistors difficult. I would suggest a nonconductive injection layer instead to see the possibility of further optimisation.

This is a good point, thanks for the comment. We opted in this case for a semiconductor since commonly adopted charge injection layers (PEI, Cs salts, etc...), which we also adopted in other cases, are not that suitable for the high temperature annealing processes we used in this work. P(NDI2ODT2) was selected because it typically leads to reasonably good electron injection properties when in contact with gold electrodes (*Appl. Phys. Lett.* **96**, 183303-183303, 2010). Moreover, it easily allows to be processed as an extremely thin film (a monolayer in this case, where we employed spin-coating), and when annealed at 310 °C is robust to the subsequent deposition of another polymer layer in toluene.

Regarding its possible contribution to the overall transport, we cannot expect an effect besides injection, since electron mobility in such a spin-coated monolayer of P(NDI2ODT2) is only 0.01 cm²/Vs, therefore more than two orders of magnitude lower than in the case of PThDPPTf4 FETs. A comparison between the electrical characteristics of the P(NDI2ODT2) monolayer only FET and PThDPPTf4 FETs with and without interlayer is reported in Figure R3 below.

Figure R3. a) sketch of FET including an ultra-thin P(NDI2ODT2) electron injection layer; b) AFM image of ultra-thin P(NDI2ODT2) injection layer; c) transfer curves of PThDPPTf4 FETs, annealed at T_2 , with (blue

dashed lines) and without (blue solid lines) the charge injection layer, and transfer curve of the injection layer only (black dash lines). In panel c), red lines indicate gate currents.

Moreover, if we compare mobility vs. temperature data for PThDPPTThF4 FETs with and without the charge injection layer P(NDI2ODT2) (Figure R4 below), we note that besides negligible differences in the mobility values, the temperature independence close to room temperature, and the thermal activation below 280 K is perfectly confirmed, see figure below.

Figure R4. Mobility vs. temperature for PThDPPTThF4 FETs with (pink squares) and without (blue squares) the charge injection layer P(NDI2ODT2).

To clarify these points, we have added Figure R3 in the SI, as Figure S27 (Section 2.11), and we have included the following description of the injection layer deposition process in the Methods (SI):

“A charge injection interlayer was introduced in some devices, by spin-coating a 0.5 g/l P(NDI2ODT2) solution in toluene on top of the source and drain electrodes (1000 rpm, 30 sec) resulting in 2.5 nm ultra-thin films. The interlayer was annealed at 310 °C for 1 h in order to make it insoluble during the successive deposition of PThDPPTThF4 from toluene”.

4. Page 13 line 379 and 380, the number of activation energy should be 61 meV instead of 66 meV according to Figure S26 and Table S4.

Thank you for spotting this error. We corrected it.

Reviewer #3

Alessandro Luzio et al. provided an interesting topic on “Microstructural control enables temperature independent electron transport in polymer transistors”. Some ideas are very meaningful such as residual energy barriers can be bypassed in the direction of molecular alignment. Chain alignment and crystal thickness can be controlled by annealing temperature before melting point.

Thanks to the reviewer for these motivating comments. We hope that answers and additional data provided below are well received by the reviewer and helpful in clarifying his/her relevant points.

However, they provided limited data for their conclusions.

1. Although authors provide the relationship between mobility and temperature in figure 7, it is really difficult to convince the band-like transporting in their polymers. Firstly, the value of mobility is stable during the changes of $1000/T$ from 3.35 (298.5K) to 3.6 (277.8K). Since the change of T is so small, it is difficult to know the trend of mobility changes. Normally, for band-like transporting behavior, there should be a max mobility peak in their curve. So if authors want to confirm the transporting mechanism, they should provide mobility values in a range of 100 K changes. Such as in *Adv. Mater.* 2016, 28, 5276–5283 from 100 K to 350 K.

We thank the Reviewer for the comment, which recalls the one from Reviewer #1. We here answer to this point by reusing in part the answer to Reviewer #1.

The Reviewer is perfectly correct in stating that in “band-like” regime an inverse temperature dependence is observed, as in *Adv. Mater.* 2016, 28, 5276–5283.

This is not the case for our devices, and therefore we are not claiming to observe a “band-like” transport. Instead, what we observe is that for specific processing conditions of the reported semiconductor, mobility becomes independent from temperature for a high enough temperature. Our claim therefore is that close to room temperature, we are able to induce temperature independent transport in our field-effect devices. Indeed, we can interpret this as a transition regime from purely temperature activation to band-like, which may occur at higher temperatures. Higher temperature are not accessible, since a temperature well above room temperature may induce morphological changes. On the other hand, below a critical temperature that depends on the energetics of the specific semiconductor we adopt, the mobility is again thermally activated. The two boundaries are restricting the observed regime in which mobility is constant. We have reduced the temperature below such threshold only down to 250 K with the sole aim to evidence an Arrhenius type of thermal dependence, and a lower temperature range is not within the scope of our work.

Although the temperature range in which we report a constant mobility is limited, it is sufficient to evidence a clear deviation from a purely temperature activated regime. It is in fact simple to distinguish a temperature activated transport, even with low activation energy (as shown in Figure 7 in the manuscript), from an independent one: only in the case of annealing at T_2 mobility is constant with temperature from 300 K to 280 K. Even a small activation energy of 31 meV, as found for temperatures below 280 K, would have produced a clearly evident slope. It is important to note that we are reporting a constant mobility from 300 K to 280 K for two different devices with two different dielectrics (CYTOP and PMMA) in Figure 7.

To further support and confirm our claim, we have gathered more data on devices with aligned films annealed at T_2 . In Figure R1 below, we report two new sets of variable temperature transfer curves measured on FETs with PMMA (panel a and c). In one of the two sets we extend the temperature range to 310 K, adding one point with respect to the range in the paper reaching 300 K (panel a). As commented above, moving to higher temperatures is not viable, as hysteretic behavior starts to appear. Moreover, in Figure R1 we report also the same CYTOP data of the paper, where we added a previously acquired further point at 310 K (panel b). In panel c, the mobility values vs. temperature are shown, confirming what we have previously reported.

Therefore, the temperature independence of electron mobility in a range close to room temperature for FET based on aligned films annealed at T_2 is robust and reproducible.

Figure R1. (a-c) Square root of the saturation ($V_D = 60$ V) transfer curves at variable temperature, where the solid line are the measured data and the dashed lines are linear fits according to the gradual channel approximation. d) μ vs. $1000/T$ plot of T_2 annealed PThDPPTThF4 aligned films with different channel lengths and dielectric layers.

We have added the data reported in Figure R1 in the Supporting Information (SI) as Fig. S31, and referred to it in the manuscript referenced on page 13.

To avoid any misunderstanding, we have carefully revised expressions used throughout the main text to describe our evidences. We note that the observation of temperature independent transport close to room temperature (and not band-like) is clearly underlined in several points, e.g. as in the abstract and in the conclusions.

Nevertheless, we amended the text for clarity in several parts. In particular, on page 2, we modified the following paragraph:

“We demonstrate that the combination of uniaxial alignment and thermal annealing within the temperature range where the melting of crystals occurs gives access to efficient electron transport at the boundary between hopping, i.e. temperature-activated, and temperature-independent, i.e. band-like regimes.”

By rephrasing:

“We demonstrate that the combination of uniaxial alignment and thermal annealing within the temperature range where partial melting of crystals occurs gives access to efficient temperature-independent electron transport close to room temperature, at the boundary between temperature-activated and band-like regimes.”

2. So, it is also difficult to say “temperature independent electron transport in polymer transistors” based on present data.

Having clarified our claim and further supported its solidity, we agree that it is in any case important to avoid possible misunderstandings by modifying the title of the manuscript, specifying that temperature independence of transport, achieved by suppression of thermal barriers, is observed close to room temperature. The new title is:

“Microstructural control suppresses thermal activation of electron transport at room temperature in polymer transistors”

3. Is it better to limit the types of polymers? What is the universality of this annealing process? Is it also applicable to classical NDI based D–A materials, such as N2200? Too big a topic can easily mislead the reader.

We understand the concern of the Reviewer. We are actually studying the application of the combination of alignment and annealing tailored on the base of calorimetric data also for other polymer systems. Although such studies go beyond the present one, we can anticipate that the strategy is also effective in maximizing transport also for a naphthalene-diimide ditiophene donor-acceptor copolymer similar to P(NDI2ODT2) (or N2200).

As an example, we provide in Figure R5 here below data measured in P(NDI2DTT2) (Figure R5a). FSC data in Figure R5b allows determining the endothermal peak. We aligned polymer films by off-center spin-coating, and either measured them “as cast” (dried at 80 °C for 10 min) or annealed at a temperature within the endothermal peak (300 °C). The polarized UV-vis absorption spectra of the films are reported in Figure R5c. In the case of annealing within the endotherm, the alignment is not only preserved, but even improved,

as evident by the much stronger anisotropy. Correspondingly, from AFM analysis of the films surface, we observe a transition from a fibrillary pattern (Figure R5d) to lamellae oriented perpendicular to the backbone alignment direction (Figure R5e). This evidence is similar to what we have reported for PThDPpThF4, with the difference of increased anisotropy upon annealing within the endotherm.

Following the optical and AFM analysis, we tested the uniaxially aligned P(NDIDTT2) films in FET devices. Characteristic curves are reported in Figure R5(e-f). It is confirmed that annealing at a temperature within the endotherm maximize the transport properties. In the radial direction, FET electron mobility increases from 0.14 cm²/Vs to 0.7 cm²/Vs, and transport anisotropy increases from a factor of 20 to 39, respectively, for as cast films and films annealed at 300 °C.

We are currently extending such procedure to other D-A copolymers. We will report such results in a further comprehensive work.

Figure R5. a) Molecular structure of polymer P(NDI2DTT2). b) F-DSC data for P(NDI2DTT2). c) Polarized UV-vis spectra of uniaxially aligned P(NDI2DTT2) films through off-center spin coating, of as cast film and after annealing at a temperature within the FSC endotherm (300 °C). AFM images of P(NDI2DTT2) uniaxially aligned films, as cast (d) and after annealing at 300 °C (e); transfer curves of uniaxially aligned

P(NDIDTT2) as cast and annealed at 300 °C measured parallel (radial direction of spin) (e) and perpendicular (tangential direction of spin) (f) to the molecular alignment direction.

4. Since molecular weight is very important for polymer properties (PThDPPTThF4, M_n : 14 kg mol⁻¹; dispersity, \mathcal{D} : 3.7). Please provide the GPC data for PThDPPTThF4.

Thanks for the comment. We agree that additional data should be reported. Here below the size exclusion chromatography curve of the PThDPPTThF4 with molecular weights $M_n/M_w = 14.1/52.2$ kg/mol, measured in CHCl₃ at room temperature is reported (Figure R7). We have added such graph in the SI, Section 2.1, Figure S1.

Figure R6. SEC curve of the herein investigated, homocoupling-free PThDPPTThF4 sample with molecular weights $M_n/M_w = 14.1/52.2$ kg/mol measured in CHCl₃ at room temperature.

How does the molecular weight effect on their FET mobility?

This is a most important point, which is under current investigation. Regarding mere values of mobility as a function of molecular weight, the current study represents some of the best values. We have been trying other molecular weights of PThDPPTThF4 that are smaller and larger than the moderate molecular weight of 14 kg/mol (M_n), and mobilities were either lower or similar. A more detailed study this is certainly beyond the scope of the paper, which already contains a major data set. As mentioned, usage of further materials and molecular weights is postponed to forthcoming papers. As an example, below we propose a part of data which we already have in our hands on this specific topic.

First, it is important to verify the thermal behavior of samples depending on the molecular weight. In the SI we previously reported the FSC data for different molecular weights. In Figure S6 we reported the thermal history of PThDPPTThF4 samples of 11, 14 and 30 kg/mol (M_n). The molecular weight is found to determine the position of the endothermic peaks, which increases with increasing molecular weight. By carefully tuning T_2 (the T_2 applied for the 11, 14 and 30 kg/mol PThDPPTThF4 samples were 250, 287 and 290 °C, respectively) the effect of thermal annealing at T_2 produces the same effect in the thermal scans of all

samples. This confirms that intense lamellar thickening processes occur in all the materials systems subject to annealing stages at T_2 .

Then, we verified the effect of molecular weight on FET mobility and its dependence on thermal annealing (not reported in the current study). In this case we selected three batches of PThDPPTThF4 with different M_n (11, 20 and 30 kg/mol) with respect to the one adopted for the main manuscript (14 kg/mol). In the Figure R7 below it is possible to see that for low annealing temperature (equivalent to T_1), the mobility trend with M_n is very clear: very low mobility is achieved with low M_n , two orders of magnitude higher mobility is achieved with the highest M_n , and an intermediate mobility value is achieved for intermediate M_n . The second aspect we can note is that an increase in mobility is achieved in all samples with increasing temperature: interestingly a boost is noticeable for annealing temperatures within the respective endotherm.

We note that for M_n equal or above 14 kg/mol, upon annealing within the endotherm, a mobility comprised in the 1 to 3 cm^2/Vs range is achieved.

Figure R7. Mobility vs. annealing temperature of PThDPPTThF4 samples with varying M_n (entry 1: $M_n / M_w = 11/16$ kg/mol; entry 2: $M_n / M_w = 22/55$ kg/mol; entry 3: $M_n / M_w = 30/72$ kg/mol). The correspondent endotherm of first heating DSC scan is also reported. DSC data corresponds quite well with FSC data, already reported in SI (Figure S6).

The detail molecular structure is lacking for PThDPPTThF4, such as what is kind of R group?

The structure of PThDPPTThF4 is reported in Figure 1, where in the caption it is indicated the nature of the R group: 2-octyldodecyl.

Some details for FET fabrication are missing, such as output curves of their devices;

We agree output characteristics curve are to be reported. We included Figures R8 and R9 here below in the SI, Section 2.10, as Figure S25 and S26, respectively. We have indicated in the manuscript the presence of output curves in the SI.

Figure R8. Transfer characteristic ($V_{DS} = 60$ V) and output curves of PThDPPTf4 FETs based on uniaxially aligned films annealed at T_1 ($W = 2000$ μm , $L = 100$ μm).

Figure R9. Transfer characteristic ($V_{DS} = 60$ V) and output curves of PThDPPTf4 FETs based on uniaxially aligned films annealed at T_2 ($W = 1000$ μm , $L = 80$ μm).

calculating equation of effective mobility;

We calculated the effective mobility μ_{eff} in order to demonstrate the robustness of our mobility data. The definition is reported in the paper by Choi et al. Nat Mater 2017, 17, 2. It can be calculated by multiplying the apparent mobility μ , obtained from the local slope of the transfer curve (as described in the Methods), by the measurement reliability factor r : $\mu_{\text{eff}} = r \times \mu$. The equations for the calculation of r are reported in Nat Mater 2017, 17, 2.

We have added to the Methods in the SI the following statement:

“The effective mobility μ_{eff} was obtained by multiplying μ_{sat} by the measurement reliability factor r : $\mu_{\text{eff}} = r \times \mu_{\text{sat}}$. r was determined according to the definition and equation reported in the work by Choi et al.^{[2]”}

how to prepare an interlayer of P(NDI2ODT2) on electrode (Figure S25), does it not destroy during PThDPPTfF4 coating?

This is a good point, and echoes what asked by Reviewer #2. We answer here to this Reviewer by adapting what already answered to Reviewer #2.

First, P(NDI2ODT2) was selected because it typically leads to reasonably good electron injection properties when in contact with gold electrodes (*Appl. Phys. Lett.* **96**, 183303-183303, 2010). Moreover, it easily allows to be processed as an extremely thin film (a monolayer in this case, where we employed spin-coating), and when annealed at 310 °C is robust to the subsequent deposition of another polymer layer in toluene. Such annealing process is key to avoid it is dissolved during PThDPPTfF4 coating.

To better describe the injection layer, we have added Figure R3 in the SI, as Figure S27, and we have included the following description of the injection layer deposition process in the Methods (SI):

“A charge injection interlayer was introduced in some devices, by spin-coating a 0.5 g/l P(NDI2ODT2) solution in toluene on top of the source and drain electrodes (1000 rpm, 30 sec) resulting in 2.5 nm thick ultra-thin films. The interlayer was annealed at 310 °C for 1 h in order to make it insoluble during the successive deposition of PThDPPTfF4 from toluene”.

Figure R3. a) sketch of FET including an ultra-thin P(NDI2ODT2) electron injection layer; b) AFM image of ultra-thin P(NDI2ODT2) injection layer; c) transfer curves of PThDPPTfF4 FETs, annealed at T_2 , with (blue dashed lines) and without (blue solid lines) the charge injection layer, and transfer curve of the injection

layer only (black dash lines). In panel c), red lines indicate gate currents; d) saturation mobility vs. V_{GS} of PThDPThF4 FETs, annealed at T_2 , with (blue dashed lines) and without (blue solid lines) the charge injection layer.

In main text, figure S26 is for PThDPThF4 or N2200?

Thanks for asking to clarify. In the former Figure S26 (Figure S30 in the revised version), panel a) refers only to PThDPThF4, where we highlight how the activation energy is gate voltage dependent in the transport direction parallel to backbone alignments, while it is higher and independent from the gate voltage in the orthogonal direction. In panel b) instead we compare mobility vs. temperature data measured on PThDPThF4 FET (taken from Figure 7 in the main text) with those measured on N2200 (or P(NDI2ODT2)). As this was not clear, we amended the figure caption.

Reviewers' comments:

Reviewer #1 (Remarks to the Author):

The authors have changed the key message of the work from band-like transport to "temperature independent transport". I agree with such approach and the results confirm that in the investigated temp. range the charge carrier mobility is independent from temperature. However, the temperature range is very narrow. I think that it is important to provide a deeper discussion how these results correspond to recent literature. Most publications distinguish between thermally activated and band-like (or band) transport. The question is if such intermediate region (thermally independent, but not yet band-like transport) has been observed or theoretically suggested before? I am not fully convinced about the importance of these findings, especially taking into account publication of these results in a high-ranking journal. If band-like transport is not the case (like answered by authors), the novelty and importance of the work should be proved by the authors.

Reviewer #2 (Remarks to the Author):

I have carefully checked the response from the authors to my questions and the comments and questions of other reviewers. I think the authors have well answered and addressed all the comments and questions. I thereby recommend acceptance of the manuscript without further revision.

Reviewer #3 (Remarks to the Author):

I think the authors almost addressed the most of my concerns. But there still some a few minor points need to be revised.

1. Since the active layers in their devices are fabricated from temperature higher than 80 °C, it is really difficult to say temperature higher than 300K (27 °C) is higher. So the claim "Higher temperature are not accessible, since a temperature well above room temperature may induce morphological changes" may be not correct.
2. Some descriptions are not scientific enough, it is better to give a precise description, such as "a range close to room temperature" "temperature independent transport close to room temperature". It is difficult to understand "close to room temperature".
3. The unit of temperature was misusing in Figure R1a and R1d. It should be "K" not "°C". So it's better to unify units in the whole text.

General Remarks

We thank again the reviewers for further considering our revised work and for a constructive collaboration to improve our work. We are pleased that we have been able to clarify our findings. We have addressed in detail here below the remaining concerns.

Reviewer #1

The authors have changed the key message of the work from band-like transport to "temperature independent transport". I agree with such approach and the results confirm that in the investigated temp. range the charge carrier mobility is independent from temperature.

We are pleased that we have been able to show the solidity of our data. We would like to underline that we have not changed the key message though. The temperature independence of mobility was the main claim from the first version of our manuscript. We acknowledge that in the original version part of the discussion could have been misinterpreted, but our claims never went beyond the experimental observation.

As an example, to avoid confusion, we have changed the title from the original version to the first revised version:

“Microstructural control enables temperature independent electron transport in polymer transistors”

to

“Microstructural control suppresses thermal activation of electron transport at room temperature in polymer transistors”

The same concept was present from the first version of the abstract as well “Here we show that precise processing of an [...] electron transporting copolymer results in highly reliable, single crystal-like and voltage-independent mobility with vanishing activation energy close to room temperature.” The same applies to the claims in the original and present conclusions:

“We have shown that combined uniaxial alignment and rationally selected thermal annealing procedures of films of PThDPPTThF4 strongly enhance field-effect transistor properties, culminating in a thermally independent transport regime close to room temperature”

and

“Owing to the retention of backbone orientation upon annealing at T2, we were able to observe that energetic barriers for charge transfer are strongly reduced exclusively in the direction of chain alignment, leading to temperature-independent electron transport near room temperature”

And

“In conclusion, for a low-molecular weight, high performance electron transporting copolymer, synthesized free of homocoupling defects by a simple direct arylation polycondensation protocol, we have identified key parameters required to induce a transition from thermally activated to temperature independent and charge density independent, single crystal like electron transport”

None of the claims referred to band-like transport.

However, the temperature range is very narrow. I think that it is important to provide a deeper discussion how these results correspond to recent literature. Most publications distinguish between thermally activated and band-like (or band) transport. The question is if such intermediate region (thermally independent, but not yet band-like transport) has been observed or theoretically suggested before? I am not fully convinced about the importance of these findings, especially taking into account publication of these results in a high-ranking journal. If band-like transport is not the case (like answered by authors), the novelty and importance

of the work should be proved by the authors.

As stated in the manuscript, inverted temperature dependence of mobility, or band-like transport, in polymer transistors have been observed so far in only four cases for p-type devices (Y. Yamashita, J. Takeya et al., *Adv. Mater.* 2014, 26, 8169–8173; Y. Yamashita, J. Takeya et al., *Chem. Mater.* 2016, 28, 420–424; S. Schott, H. Sirringhaus et al., *Adv. Mater.* 2015, 27, 7356–7364; J. Lee, M.-S. Kang et al., *J. Am. Chem. Soc.* 2015, 137, 7990–7993) and in one case only for n-type devices (S. P. Senanayak, S. Patil, K. S. Narayan et al., *Phys. Rev. B* 2015, 91, 115302).

The temperature range in which inverted temperature activation has been observed in such cases is *always similarly limited*, typically from 270 K to 320 K. Moreover, in all cases, the inverted temperature dependence is very mild, accounting in the best case for 50 % variation in mobility with increasing the temperature, and it is anticipated by a range in which mobility is basically constant. Within the same data, cases where mobility is substantially flat with temperature can be found: the mobility vs. T plot at 40 V gate voltage in *Phys. Rev. B* 2015, 91, 115302; the one for a spin-coated sample in the above mentioned reference *Adv. Mater.* 2015, 27, 7356–7364. Therefore such intermediate region has been found in works reporting “band-like” transport.

We also would like to highlight that we have been very careful in reporting the mobility vs. temperature plot. As largely documented in our work, we have obtained very reliable mobility data by using long channels (80 – 100 μm). With shorter channels (20 μm) that give rise to strong gate voltage dependence in mobility, we could observe mild inverted temperature activation (as reported in the plot below). Since this behavior could be affected by the non ideality of the device, we avoided to claim the observation of band-like and restricted our claim to temperature independent mobility, which is supported by solid mobility data.

Overall, we have reported solid and extensive data of a rare case of clear deviation from temperature activation of electron transport in polymer FETs, and we have linked this to the microstructural evolution of films thanks to advanced and detailed investigations. This allowed us to highlight for the first time the key elements for the suppression of energetic barriers to transport in polymer films, thus delivering important guidelines to the community for further development in the direction of high-performance polymer FETs characterized by a transport regime beyond temperature activated ones. We anticipate these results will pave the way to further improvement and stimulation of the field to foster practical applications.

We hope to have answered in sufficient detail to this point raised by the referee.

Reviewer #2

I have carefully checked the response from the authors to my questions and the comments and questions of other reviewers. I think the authors have well answered and addressed all the comments and questions. I thereby recommend acceptance of the manuscript without further revision.

We thank the reviewer for the positive and motivating feedback.

Reviewer #3

I think the authors almost addressed the most of my concerns.

We are pleased we have been able to address the main concerns of the reviewer and clarify our findings.

But there still some a few minor points need to be revised.

1. Since the active layers in their devices are fabricated from temperature higher than 80 oC, it is really difficult to say temperature higher than 300K (27 oC) is higher. So the claim “Higher temperature are not accessible, since a temperature well above room temperature may induce morphological changes” may be not correct.

We understand the point made by the reviewer. Such statement, which we wrote in the previous rebuttal, refers to the accessible temperature range of electrical analysis within which we obtain reliable data. We do adopt much higher temperatures to process films, but when performing electrical characterization, we have to restrict to the range in which ideality of the transistor characteristics are obtained, e.g. avoiding to reach temperatures that triggers hysteresis.

In this case, we are restricted to temperatures near 300 K, since above this temperature electrical characteristics of the devices are no longer ideal as reported in the manuscript, making the extraction of mobility, and therefore determination of its thermal dependence, quite unreliable. The first effect we notice above 300 K is the appearance of hysteresis, as it can be seen in panel b) here below. We do not have an explanation for this effect yet.

In the manuscript we therefore restricted the temperature range to 300 K. In the SI we also added the data at 310 K to show that mobility keeps constant, since the extraction of mobility is still reasonable given the mild non ideality.

2. *Some descriptions are not scientific enough, it is better to give a precise description, such as “a range close to room temperature” “temperature independent transport close to room temperature”. It is difficult to understand “close to room temperature”*

We agree with the referee that such expressions are not very specific. We have modified correspondingly the text, as follows.

- Abstract: “with vanishing activation energy close to room temperature” replaced with “with vanishing activation energy above 280 K”.
- Page 2: “gives access to efficient temperature-independent electron transport close to room temperature” replaced with “gives access to efficient temperature-independent electron transport above 280 K”.
- Conclusions: “culminating in a thermally independent transport regime close to room temperature” replaced with “culminating in a thermally independent transport regime in the proximity of 300 K, i.e. close to room temperature.”

and

“leading to temperature-independent electron transport near room temperature” replaced with “leading to temperature-independent electron transport near to 300 K”.

3. *The unit of temperature was misusing in Figure R1a and R1d. It should be “K”not “oC”. So it's better to unify units in the whole text.*

Thank you for spotting this error. We have corrected Figure S31 (Figure R1 of the previous rebuttal letter).

In the main text we preferred to keep different units for temperature, using °C for processing temperature (FSC, annealing temperatures) and K for variable temperature measurements, as this is common practice in the reference literature.

REVIEWERS' COMMENTS:

Reviewer #1 (Remarks to the Author):

The manuscript can be now published.

Reviewer #3 (Remarks to the Author):

I think the authors have well addressed my comments. I thereby recommend acceptance of the manuscript without further revision.